# Wheat *Pm55* alleles exhibit distinct interactions with an inhibitor to cause different powdery mildew resistance

Chuntian Lu[1,2,9], Jie Du[1,2,9], Heyu Chen[1,2], Shuangjun Gong[3], Yinyu Jin[1,2], Xiangru Meng[1,2], Ting Zhang[1,2], Bisheng Fu[2,4], István Molnár[5,6], Kateřina Holušová[6], Mahmoud Said [6,7], Liping Xing[1,2], Lingna Kong[1], Jaroslav Doležel [6], Genying Li[8], Jizhong Wu[2,4], Peidu Chen[1], Aizhong Cao[1,2] & Ruiqi Zhang [1,2] ✉

Powdery mildew poses a significant threat to wheat crops worldwide, emphasizing the need for durable disease control strategies. The wheat-*Dasypyrum villosum* T5AL·5 V#4 S and T5DL·5 V#4 S translocation lines carrying powdery mildew resistant gene *Pm55* shows developmental-stage and tissue-specific resistance, whereas T5DL·5 V#5 S line carrying *Pm5V* confers resistance at all stages. Here, we clone *Pm55* and *Pm5V*, and reveal that they are allelic and renamed as *Pm55a* and *Pm55b*, respectively. The two *Pm55* alleles encode coiled-coil, nucleotide-binding site-leucine-rich repeat (CNL) proteins, conferring broad-spectrum resistance to powdery mildew. However, they interact differently with a linked inhibitor gene, *SuPm55* to cause different resistance to wheat powdery mildew. Notably, *Pm55* and *SuPm55* encode unrelated CNL proteins, and the inactivation of *SuPm55* significantly reduces plant fitness. Combining *SuPm55/Pm55a* and *Pm55b* in wheat does not result in allele suppression or yield penalty. Our results provide not only insights into the suppression of resistance in wheat, but also a strategy for breeding durable resistance.

Host resistance is the most effective way to reduce yield loss caused by crop diseases. Unfortunately, deployed resistance (R) genes frequently become ineffective due to genetic changes in the pathogens. Breeding for resistance is costly and introduction of resistance genes may reduce the rate of yield increase[1]. Thus, breeding varieties with durable resistance without compromising yield potential is a major challenge in crop improvement.

Plants and pathogens have formed various interactions during the long-term coevolution. Many host plants exhibit resistance during the entire life cycle often referred to as seedling or all-stage resistance (ASR)[2]. ASR genes typically encode nucleotide-binding leucine-rich repeat (NLR) receptors that recognize specific pathogen avirulence (Avr) proteins, providing race-specific resistance. Thus, such genes are generally nondurable when deployed singly[3]. Interestingly, paired

[1]College of Agronomy of Nanjing Agricultural University/State Key Laboratory of Crop Genetics & Germplasm Enhancement and Application /JCIC-MCP, Nanjing 210095, P.R. China. [2]Zhongshan Biological Breeding Laboratory, No.50 Zhongling Street, Nanjing, Jiangsu 210014, China. [3]Institute of Plant Protection and Soil Science, Hubei Academy of Agricultural Sciences, Wuhan 430064, P.R. China. [4]Institute of Germplasm Resources and Biotechnology/Jiangsu Provincial Key Laboratory of Agrobiology, Jiangsu Academy of Agricultural Sciences, Nanjing 210014, P.R. China. [5]Agricultural Institute, Centre for Agricultural Research, Eötvös Loránd Research Network (ELKH), 2462 Martonvásár, Hungary. [6]Institute of Experimental Botany of the Czech Academy of Sciences, Centre of Plant Structural and Functional Genomics, Šlechtitelů 31CZ 77900 Olomouc, Czech Republic. [7]Field Crops Research Institute, Agricultural Research Centre, 9 Gamma Street, 12619 Giza, Cairo, Egypt. [8]Crop Research Institute, Shandong Academy of Agricultural Sciences, Jinan 250100, P.R. China. [9]These authors contributed equally: Chuntian Lu, Jie Du. ✉e-mail: zrq@njau.edu.cn

NLRs, such as PigmS/PigmR, have been shown to confer durable resistance against rice blast[4]. In other cases, resistance is restricted to particular developmental stages, the best known of which is adult plant resistance (APR)[5]. Some APR genes encode non-NLR receptors and confer race non-specific and durable resistance[6]. The examples include *Yr36* which is a cytoplasmic protein kinase gene[7], *Lr34* is an ABC transporter gene and *Lr67* is a hexose transporter gene[8,9]. Nevertheless, certain well-documented APR genes of wheat, such as *Lr12*, *Lr37* and *Yr49*, still confer race-specific resistance[10]. Notably, rice adult-plant race-specific genes *Xa3* and *Xa4* for resistance to bacterial blight demonstrated durability[11]. Additionally, host plants may also express tissue-specific resistance[12,13]. For instance, some wheat genotypes with either ASR or APR to rust may develop susceptible symptoms on spikes. Consequently, despite numerous NLR genes and atypical resistance genes, such as receptor-like kinases (RLKs), tandem kinase proteins (TKPs), like receptor-like kinases (WAKs) and transcription factors (TFs), have been isolated from plants in the past decades, our understanding of the molecular mechanisms underlying divergent resistance among host plants remains limited.

NLR encoding locus that contains a single R gene with one or more alleles encoding different resistance specificities is common in plant kingdom. For examples, wheat powdery mildew (Pm) resistance gene *Pm2* includes eight functional alleles[14], and *Pm3* includes 17 functional alleles identified to date[15]. Recent studies have expanded our understanding of the avirulence (AVR) effectors from the pathogen recognized by allelic series, which were sequence unrelated proteins[16,17]. Thus, as an alternative to the pyramiding of different R genes, different allelic variants can also be combined for achieving more durable and broad-spectrum resistance[18]. However, in some cases, pyramiding different R genes or alleles can cause mutual suppression of their functions. A few reports have described the genetic basis of disease resistance suppression in wheat with the examples including suppression of stem rust resistance by a subunit of the mediator complex gene[19], and direct interactions between homologous NLR receptors encoded by alleles of *Pm3* and *Pm8* to suppress powdery mildew resistance[20,21]. In contrast, our knowledge regarding the suppression of unrelated NLR immune receptors is limited.

Powdery mildew, caused by *Blumeria graminis* f. sp. *tritici* (*Bgt*), is a common disease that limits wheat production in temperate regions. To enhance host resistance, numerous disease resistance genes have been transferred into bread wheat from its wild relatives. Remarkably, five powdery mildew resistance genes, *Pm21*, *Pm55*, *Pm62*, *Pm67* and *Pm5V* have been introgressed from different *Dasypyrum villosum* (L.) (2n = 2x = 14, VV) accessions into wheat through Robertsonian translocations (RobTs)[22–24]. Among these, wheat-*D. villosum* T5DL·5 V#4 S and T5AL·5 V#4 S translocation lines, carrying the gene *Pm55* from *D. villosum* accession 91C43, showed developmental-stage and tissue-specific resistance to wheat powdery mildew[12]. However, the T5DL·5 V#5 S translocation line, carrying the gene *Pm5V* from *D. villosum* accession 01I140, exhibited broad effectiveness against *Bgt* races at all stages[24]. Therefore, the different 5VS-introgression lines present an opportunity to isolate and pyramid genes that confer distinct resistance in wheat. Additionally, previous research has shown that the transfer of *Pm55* and *Pm5V* into diverse wheat genetic backgrounds did not negatively affect yield-related traits[24,25]. Consequently, these genes have been utilized in the wheat breeding program in China to develop elite resistance lines[26].

In this work, we aim to elucidate the genetic and molecular basis underlying the distinct types of resistance against powdery mildew as observed in different 5VS-translocation lines. Our findings reveal that *Pm55* and *Pm5V* are allelic on chromosome arm 5VS; however, they have divergent interactions with a linked inhibitor gene, *SuPm55*, which causes the distinct resistance of different 5VS-translocation lines. Notably, *Pm55* and *SuPm55* are unrelated NLR proteins and knockout of *SuPm55* significantly reduces the plant fitness.

Accordingly, our results reveal the complex interactions of different NLR immune receptors in disease resistance, and provide insights into the suppression of resistance in wheat. Combining the T5AL·5 V#4 S and T5DL·5 V#5 S translocations offers a promising strategy for breeding durable resistance to wheat powdery mildew.

## Results

### High-resolution mapping reveals an inhibitor linked with *Pm55*

To accurately evaluate the response of the 5VS-translocation lines to powdery mildew, the translocated chromosomes T5AL·5 V#4 S, T5DL·5 V#4 S and T5DL·5 V#5 S were transferred into a highly susceptible cv. NAU0686 genetic background, respectively. Subsequently, stable translocation lines were developed in the BC$_5$F$_6$ progeny (Supplementary Table 1). Upon inoculation of these lines with *Bgt* isolate E09, we observed that both T5AL·5 V#4 S translocation line NAU185 and T5DL·5 V#4 S line TF5V-1, carrying *Pm55* displayed susceptible at 3-leaf stage (IT 4), but exhibited high resistance after the 5-leaf stage (IT 1) with susceptible leaf sheaths only at adult-plant stage (Fig. 1a, b). While, T5DL·5 V#5 S translocation line NAU1908, carrying *Pm5V* consistently demonstrated high resistance to powdery mildew across all stages and tissues (IT 0;). Trypan blue and DAB staining showed large number of spores produced in TF5V-1 seedling leaf (3-leaf) after inoculation with E09 (Supplementary Fig. 1a, c), but very mild cell death and robust accumulation of $H_2O_2$ in NAU1908 seedling leaf (Supplementary Fig. 1b, d). Collectively, chromosome arms 5 V#4 S and 5 V#5 S conferred distinct forms against powdery mildew in wheat.

To map *Pm55* and *Pm5V*, a genetic mapping population containing 5425 F$_2$ individuals was previously constructed by crossing two T5DL·5VS translocation lines, TF5V-1 and NAU1908[24]. *Pm5V* in NAU1908 was fine mapped to an approximately 0.9 Mb region referencing the genome sequence of wheat cv. Chinese Spring 5DS using forty-six crossovers on chromosome arm 5VS[24]. However, it was unclear whether *Pm55* and *Pm5V* are allelic. To test their allelism, we further inoculated the F$_3$ and F$_4$ homozygous recombinant lines with *Bgt* isolates E09, E26 and E31 at seedling stage, and E09 at adult plant stage. As well as, sixty-three InDel markers and six EST-STS markers were used to screen these lines (Supplementary Fig. 2, Supplementary Data 1). Among these forty-six lines, we identified twenty recombinant lines between makers *SCA29236* and *SCA12100* (Type I) with the same phenotype as NAU1908, showing all-stage resistance to the three *Bgt* isolates. *Pm5V* locus was further mapped within the interval flanked by InDel markers *SCA93008* and *SCA2324*, corresponding to ~140 kb on the sequenced *D. villosum* 91C43$^{DH}$ genome[27]. Additionally, we observed another set of twenty recombinant lines between makers *SCA10690* and *SCA25706* (typeII) with phenotypes identical to parent TF5V-1, showing susceptibility to all three *Bgt* isolates at the seedling stage but resistance to E09 at the adult plant stage, with lower susceptible sheaths. This suggested that *Pm55* is mapped within the interval flanked by InDel markers *SCA11392* and *SCA23759*. However, the remaining six recombinants between makers *SCA11392* and *SCA93008* (R5VS-13 to R5VS-18, type III) had contrasting phenotypes. These lines exhibited resistance to E09 and E26 but susceptibility to E31 at the seedling stage, and resistance to E09 at adult plant stage without susceptible leaf sheaths, which was distinct from the parental phenotypes of TF5V-1 and NAU1908. Based on the genotypes of type II and type III lines, one possibility is another resistance locus on 5V#5 S, flanked by InDel markers *SCA11392* and *SCA77919*, could provide the seedling resistance to E09 and E26 in type III recombination lines, or that another possibility is an inhibitor in the *Pm55* interval suppressed *Pm55*-mediated resistance to E09 and E26 in type II recombination lines. Notably, none of the recombinants was susceptible to E09 at the adult plant stage, suggesting that *Pm55* and *Pm5V* could indeed be allelic or close linkage.

To fine map the resistance or suppression locus linked with *Pm55*/*Pm5V* on 5VS, a cross was made between TF5V-1 and the Type III

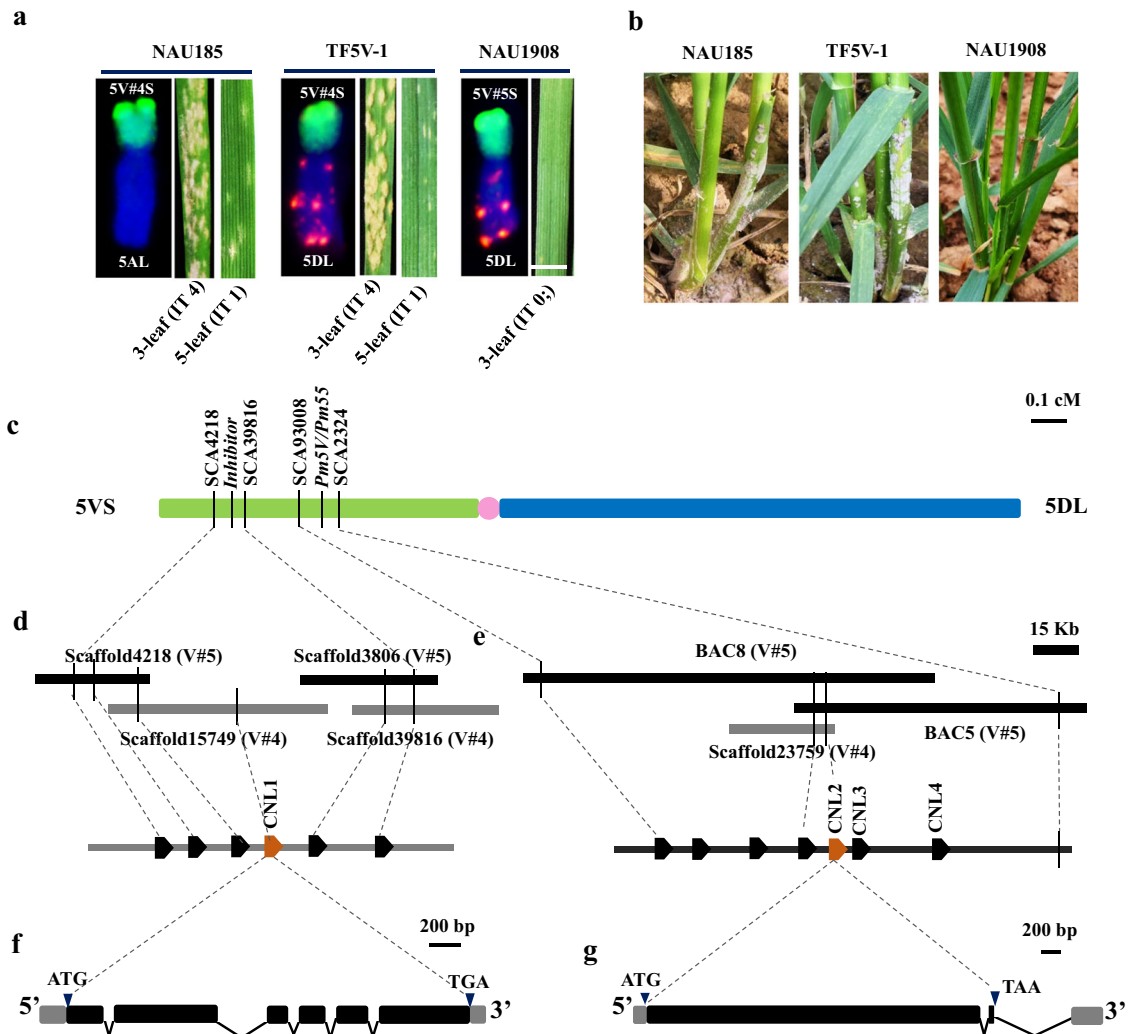

**Fig. 1 | Map-based cloning of the candidates in both inhibitor and *Pm55/Pm5V* intervals on the chromosome arm 5VS of *D. villosum*. a** Seedling responses of three 5VS-introgression lines inoculated with *Bgt* isolate E09. At least ten plants of each lines were observed. The translocated chromosomes in each lines were identified by GISH/FISH, among them, *D. villosum* gDNA labeled with fuorescein-12-dUTP (green) as probe was used for GISH, and FISH probe was Oligo-pAs1 labeled with TAM (red) to produce the red signals on the chromosomes of wheat D genome. Scale bar, 5.0 mm. **b** Response to powdery mildew of 5VS-introgression lines at adult-plant stage. **c** Fine mapping *Pm55/Pm5V* and inhibitor intervals on chromosome arm 5VS. **d** Physical maps with four overlapping 5VS scaffolds, annotating six genes in the inhibitor interval. **e** Physical maps with two overlapping BACs, annotating seven genes in the *Pm55/Pm5V* interval. **f** Gene structure of *CNL1* in TF5V-1. **g** Gene structure of *CNL2* in NAU1908. Black rectangles, black lines and gray rectangles indicate exons, introns and UTRs respectively.

---

recombination line R5VS-15 (Supplementary Fig. 3). All $F_1$ seedlings were susceptible to E09 but showed adult-plant resistance with lower sheaths susceptibility, similar to TF5V-1. The $F_2$ population segregated into 128 resistant and 394 susceptible seedlings (Supplementary Table 2). However, all the $F_2$ plants showed adult-plant resistance, but the seedling susceptible plants of them had lower sheaths susceptibility. These results indicated that there was not a resistance gene on 5V#5 S, but a dominant inhibitor linked with *Pm55* on 5V#4 S, suppressing *Pm55*-mediated resistance on seedlings and adult-plant leaf sheaths. Further genotyping of 3882 $F_2$ individuals identified six crossovers between markers *SCA11392* and *SCA77919*. Combination with five newly developed markers, the suppression interval was narrowed down to a ~100 kb region flanked by InDel markers *SCA4218* and *SCA39816* by comparison with the *D. villosum* 91C43$^{DH}$ genome (Supplementary Fig. 4, Fig. 1c and Supplementary Data 2). In summary, a single gene *Pm5V* in NAU1908 conferred resistance to powdery mildew at all stages and tissues, whereas *Pm55* together with a linked inhibitor in TF5V-1 conferred the developmental-stage and tissue-specific resistance.

## Dissection of the candidates for both intervals

Diploid *D. villosum* is an outcrossing species that may produce progenies with complex haplotypes in genome locus of interest. Thus, DNA sequences of chromosome arms 5V#4 S and 5 V#5 S are necessary to isolate the haplotypes in the aimed loci on them. Previously, we flow-sorted and sequenced chromosome arm 5 V#5 S, resulting in 173.4 Mb sequence with an average scaffold N50 size of 22.4 kb[24]. Building on this, we further isolated T5DL·5 V#4 S translocated chromosome from TF5V-1 by flow sorting and then sequenced it using Illumina short read technology (Supplementary Fig. 5a, b). The size of the 5V#4 S de novo assembly was 171.8 Mb with an average scaffold N50 size of 18.3 kb, including 1262 high-confidence protein-coding genes annotated using protein homology-based prediction methods.

To isolate inhibitor gene, the inhibitor physical region between markers *SCA4218* and *SCA39816* on 5 VS of 91C43$^{DH}$ was first subjected to collinearity analysis with Chinese Spring 5AS, 5BS and 5DS, revealing that the annotated genes in this region across different genomes were highly conserved (Supplementary Fig. 5c). Subsequently, we aligned the homologies of annotated genes between 5V#4 S and 5 V#5 S

scaffolds and the inhibitor physical region on different genomes. We identified two 5 V#5 S scaffolds, Scaffold4218 (20,202 bp) and Scaffold3806 (39,439 bp), and two 5 V#4 S scaffolds, Scaffold15749 (54,335 bp) and Scaffold38916 (46,667 bp), containing homologs of the genes located in the inhibitor region of 5AS, 5BS and 5DS, which generated a contiguous 100,963 bp sequence between markers *SCA4218* and *SCA39816* (Supplementary Fig. 5d). This combination sequence included six annotated genes. Among them, an intact coiled-coil nucleotide-binding leucine-rich repeat (CNL) gene, *CNL1* (G4), along with gene G3 were absent in the inhibitor region on 5 VS of 91C43[DH] (Fig. 1d, Supplementary Fig. 5c). Full-length transcriptome sequencing of TF5V-1 seedling leaves inoculating with *Bgt* isolate E09 revealed that *CNL1* was the only intact transcribed gene of the six annotated genes, and therefore could be the candidate of a functional gene. The full genomic sequence of *CNL1* in 5V#4 S was further confirmed by PCR, which spanned 2651 base pairs (bp) from ATG start codon to TGA stop codon with six exons and five introns, and generating a predicted 565 amino-acid protein (Fig. 1f).

However, the 5 V#4 S and 5 V#5 S scaffolds containing the homologs of annotated genes on the genomes of CS 5AS, 5BS and 5DS between *Pm55/Pm5V* linked markers *SCA93OO8* and *SCA2324* did not cover the *Pm55/Pm5V* interval. Thus, we constructed a bacterial artificial chromosome (BAC) library using DNA from *D. villosum* 01I140 (the donor of 5V#5 S) for long-read sequencing. Screening the BAC library with markers *SCA93OO8* and *SCA2324* enabled us to identify and sequence ten BACs. Out of them, BAC5 and BAC8 overlapped the resistance region forming a contiguous 134,990 bp sequence that contained seven annotated genes, including three CNL paralogs *CNL2*, *CNL3* and *CNL4* (Fig. 1e). *CNL2* and *CNL4* were intact genes, but *CNL3* was a pseudogene due to a disrupted open reading frame (ORF). *CNL2* and *CNL3* were highly similar (94.8% identity) whereas the similarity of *CNL2* and *CNL4* was lower (81.8% identity). Gene expression analysis revealed that only *CNL2* was induced expression in seedlings of NAU1908 after infection with *Bgt* isolate E09 (Supplementary Fig. 6a). Moreover, *Pm2-5V*, the *Pm2* ortholog on 5 VS was also used as the control to verify the expression of the *CNL2* alleles induced by *Bgt* infection. Results showed that *CNL2#4* allele transcript levels in Type III recombination line R5VS-15 were approximately eight-fold higher at 72 hpi with isolate E09 (Supplementary Fig. 6b), and *CNL2#5* transcript levels in NAU1908 were approximately three-fold higher at 24 hpi (Supplementary Fig. 6c). By contrast, the transcript level of *Pm2* orthologs, *Pm2-5V#4* and *Pm2-5V#5* did not change significantly during 0−72 hpi. Accordingly, *CNL2* alleles were prioritized as the functional gene candidates.

The full genomic sequence of *CNL2* allele in NAU1908 (designed as *CNL2#5*) amplified by PCR was identical to those of the BAC sequences spanning 3887 bp from the translation initiation (ATG) to the stop (TAA) codons, including two exons and one intron and generated a predicted 1265 amino-acid protein (Fig. 1g). We identified a 5 V#4 S scaffold, Scaffold23759 (23,842 bp), which contained the full length of the *CNL2* allele (Fig. 1e). Utilizing this sequence, we amplified the full length of the *CNL2* allele in TF5V-1 (designed as *CNL2#4*), which was 3896 bp from the translation initiation (ATG) to the stop (TGA) codon, generating a predicted 1267 amino-acid protein. *CNL2#4* and *CNL2#5* shared 95.2% identity in gDNA sequence and 91.5% identity in amino acids at the protein level (Supplementary Fig. 7).

## Validation of the *CNL1* function

We used virus-induced gene silencing (VIGS) to knock down *CNL1* in TF5V-1 to verify its function. A construct targeting the CC domain of *CNL1* in TF5V-1 resulted in resistant leaves at seedling stage (Fig. 2a). A comparison of mRNA expression by qPCR in TF5V-1 leaves infected with BSMV: *CNL1* and wild-type BSMV: γ virus showed a significant decrease in expression levels of *CNL1* transcripts (Fig. 2b). These

results indicated that *CNL1* suppressed the seedling resistance of TF5V-1.

CRISPR/Cas9-mediated genome editing technology was then employed to knock out *CNL1* in TF5V-1. We used guide RNAs (gRNAs) targeting the conserved regions in CC domain of *CNL1* (Supplementary Fig. 8a), and obtained four mutant lines with 4-bp (Del5VS-1), 5-bp (Del5VS-2), 6-bp (Del5VS-3) and 27-bp (Del5VS-4) deletions in the target region as putative knockouts of the *CNL1* gene (Fig. 2c). The inoculation of *Bgt* isolate E09 showed that the amounts of fungal growth and spores on the seedling leaves of four homozygous mutants significant reduced compared with TF5V-1 (Fig. 2d,e, Supplementary Fig. 8b to f). Homozygous $T_2$ progeny from a Del5VS-4 heterozygous $T_1$ plant were all ASR to isolate E09 without lower leaf sheath susceptibility (Fig. 2f, g Supplementary Fig. 9a, b and Supplementary Table 3), confirming the role of *CNL1* as the *Pm55* suppression gene, hereafter designated as *SuPm55*.

Additionally, we found that *SuPm55* knockout plants of Del5VS-1 to Del5VS-4 created by CRISPR/Cas9 had significantly lower number of tillers compared with TF5V-1 in a powdery mildew-free field (Supplementary Fig. 10a), resulting in lower spike number per plant (Supplementary Fig. 10c), as well as lower grain yield per plant (Supplementary Fig. 10f). While, there were no significant differences in plant height (Supplementary Fig. 10b), thousand-grain weight (Supplementary Fig. 10d) and seeds per spike (Supplementary Fig. 10e). Similar results were come out in $T_2$ population derived from a Del5VS-4 heterozygous $T_1$ plant, which showed the tillers and spikes reduced in plants with 27 bp deletion of *SuPm55* compared with plants without a deletion (Supplementary Fig. 10g, h, Supplementary Fig. 11). These results indicated that the inactivation of *SuPm55* could have negative effect on plant fitness.

## Validation of the *CNL2* alleles function

To test *CNL2* alleles function by VIGS, we designed silencing constructs based on two specific targets in the NB-ARC domain in *CNL2#5*, and one target in the NB-ARC domain of *CNL2#4* (Supplementary Fig. 12a, b). Addition of the *CNL2* silencing constructs in infected NAU1908 (*Pm5V*) and Type III recombinant line R5VS-15 (*Pm55*) led to 50 to 80% reductions in transcript levels and abundant development of powdery mildew, whereas empty vector-inoculated plants remained resistant (Fig. 3a−d, Supplementary Fig. 12c to f). These results demonstrated that the *CNL2* alleles could be functional resistance genes in both NAU1908 and R5VS-15.

To further validate the function of *CNL2* alleles, the two full-length *CNL2*-derived genomic sequences with native promoters were transformed separately into susceptible wheat cv. Fielder by *Agrobacterium*-mediated transformation to determine whether the cloned *CNL2* alleles were sufficient to confer resistance (Fig. 3e). Subsequently, $T_0$ individuals were identified by PCR analysis using *CNL2*-specific markers. A total of eleven and 15 positive $T_0$ transgenic plants with *CNL2#5* and *CNL2#4*, respectively, were obtained. They displayed various levels of transgene transcription (Supplementary Figs. 13a, b and 14 a, b) and some of them were resistant to *Bgt* isolate E09 (Fig. 3f,g). $T_1$ progeny from six positive $T_0$ plants segregated 3 R: 1 S or in more complex ratios, indicative of one or more *CNL2* insertions (Supplementary Table 4). As expected, all resistant $T_1$ seedlings co-segregated with the presence of the *CNL2*-specific marker (Supplementary Figs. 13c and 14c). Homozygous $T_2$ seedlings inoculated by the other 18 *Bgt* isolates showed that *CNL2#5* transgenic line CNL2#5T_2-9 was resistant to all isolates similar as NAU1908. While, *CNL2#4* transgenic line CNL2#4T_2-3 was resistant to 15 isolates among them, which was similar as R5VS-15. By contrast, the non-transgenic cv. Fielder was susceptible to all races (Supplementary Table 5). These results demonstrated that *CNL2#4* and *CNL2#5* conferred broad-spectrum *Bgt* resistance but had distinct response spectra in wheat.

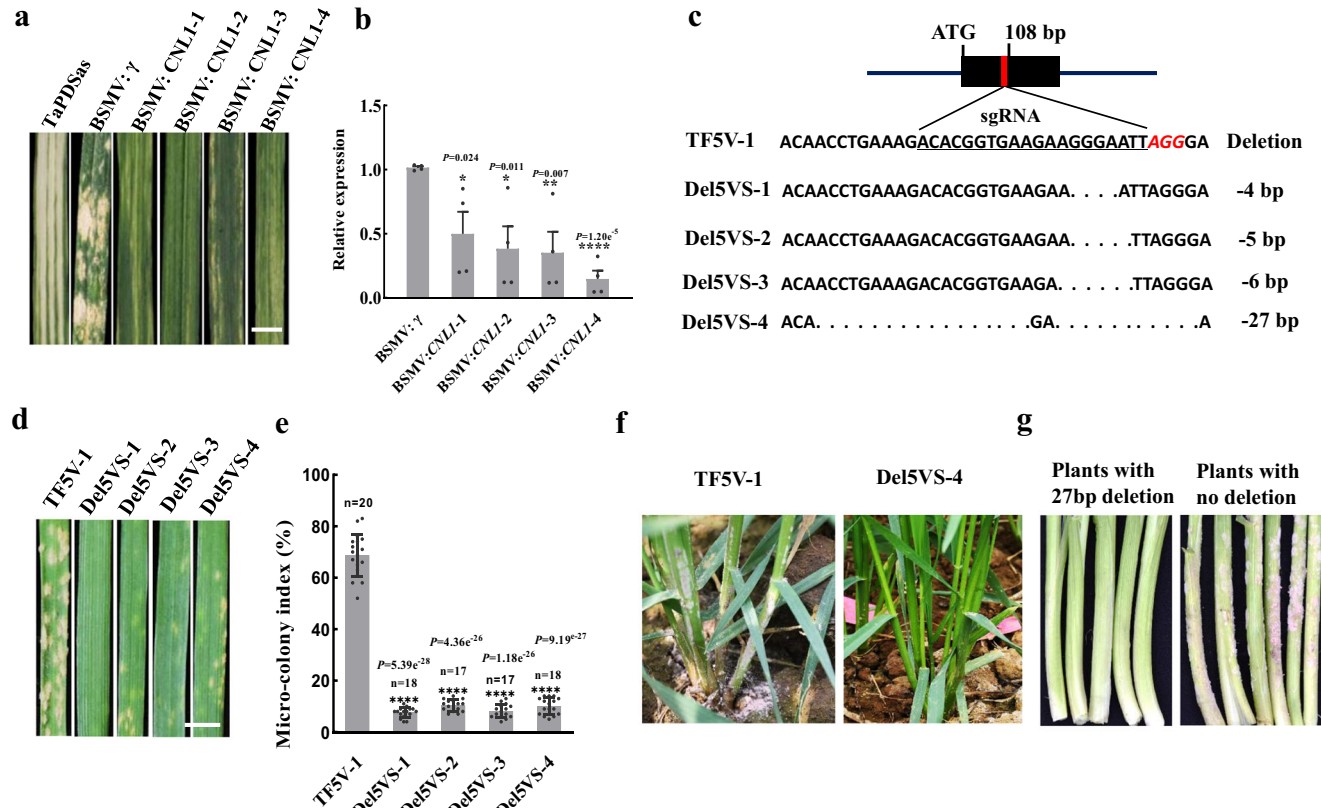

**Fig. 2 | The function validation of *CNL1* in TF5V-1. a** Target of *CNL1* CC domain silenced by VIGS in TF5V-1. *Bgt* is not visible on four independent BSMV: CNL1-infected leaves. Scale bars, 5.0 mm. **b** The expression of *CNL1* in BSMV: CNL1-infected leaves determined by qRT-PCR and compared to that in BSMV: γ leaves. Values are the mean ± SD (two-sided *t*-test, *n* = 4 biologically independent experiments, \**P* < 0.05, \*\**P* < 0.01; \*\*\*\**P* < 0.0001). **c** Four mutations in the CC domain of *CNL1* in TF5V-1 created by CRISPR/Cas9 editing. **d** Responses of CRISPR/Cas9-mediated mutant seedlings to *Bgt* isolate E09. Ten plants of each lines were recorded. Scale bars, 5.0 mm. **e** The micro-colony index decreased significantly in *CNL1* mutation seedling leaves. Value is the mean ± SD (two-sided *t*-test, \*\*\*\**P* < 0.0001). *n* = 20 in TF5V-1, *n* = 18 in Del5VS-1, *n* = 17 in Del5VS-2, *n* = 17 in Del5VS-3, and *n* = 18 in Del5VS-4. **f** The mutant Del5VS-4 showed no susceptibility on the lower leaf sheaths. **g** The plants with 27 bp deletion in the CC domain of *CNL1* showed resistance to powdery mildew on the leaf sheaths in a T₂ population, whereas plants with no deletion were leaf sheaths susceptibility. Source data are provided as a Source Data file.

To determine whether the *CNL2* alleles were required for *Pm55* and *Pm5V* resistance, we generated EMS-mutagenized populations of NAU1908 and TF5V-1. M₁ plants were inoculated with *Bgt* isolate E09. Three susceptible mutants of NAU1908 were obtained and F₁ plants from their intercrosses were also susceptible (Fig. 3h). Sequence alignments revealed a premature stop codon in the highly conserved segment of the LRR domain of *CNL2#5* in line M18 (L798Stop), and missense mutations in the conserved LRR domain in lines M477 (C1065Y) and M512 (P1089L) (Fig. 3j). In addition, ten susceptible mutants of TF5V-1 were identified in adult-plant tests (Fig. 3i). Sequencing of the *CNL2#4* in these mutants revealed SNPs causing amino acid substitutions in each mutant (Fig. 3j, Supplementary Table 6). The loss of function mutations occurring in *CNL2#4* and *CNL2#5* alleles provided additional evidence that they respectively were *Pm55* and *Pm5V*. Taken together, *Pm55* and *Pm5V* were confirmed to be functional alleles, hereafter renamed as *Pm55a* and *Pm55b*, respectively.

### Evolution and allelic variations of *SuPm55* and *Pm55*

To determine the evolutionary relationships of *SuPm55* and *Pm55* to other known R genes, we compared their protein sequences with a panel of 33 cloned NLR proteins in wheat (Supplementary Fig. 15). Their phylogenetic analysis revealed that *Pm55a* was most closely related to *Pm2* (78.1% identity), and *SuPm55* was most closely related to *Yr10* (35.8% identity). Moreover, *Pm55a* and *SuPm55* represented non-homologous CNL genes, as their DNA sequences shared very low

identity (28.8%). Comparative genomic analysis revealed that *SuPm55* homologs were absent on homoeologous group 5 chromosomes of wheat relatives rye and barley, as well as on A and D genomes of wheat, with only one copy present on the B genome. A 2875 bp deletion led to the absence of *SuPm55* alleles on 5VS genomes of NAU1908 and sequenced *D. villosum* 91C43^DH. Additionally, *SuPm55* alleles were all absent in the other five *D. villosum* accessions (Supplementary Table 1) when detection by molecular marker. Based on this, we presume that *SuPm55* could be rare in *D. villosum*.

In comparing the genomes of *Pm55* locus with that of hexaploid wheat, we revealed low sequence conservation, indicating that *Pm55* locus diverged during the evolution (Supplementary Fig. 16a). The orthologs of *CNL2*, *CNL3* and *CNL4* were all absent in wheat chromosome arm 5AS and barley 5HS, indicating that the origin of *Pm55* predates the speciation of *D. villosum* and barley, probably 14.9 million years ago[27]. Only one homolog of *Pm55* was present in each of chromosome arms 5BS and 5DS of wheat, and 5RS of rye. These homologs should be orthologous to *CNL2*, as *CNL3* was a non-complete gene, and an 84 bp sequence of *CNL4* was missing in all homologs in the sub-genome chromosome arms (Supplementary Fig. 16b). *Pm55* orthologs in wheat and its relatives had 82 to 91% identity, significantly lower than that among *Pm2* orthologs (>93% identity) (Supplementary Table 7), suggesting considerable changes after divergence of *D. villosum* and related species. The full-length sequences of *Pm55* alleles isolated from the other five *D. villosum* accessions (named haplotypes *Pm55_h1 – Pm55_h5*) differed (Supplementary Table 8), sharing

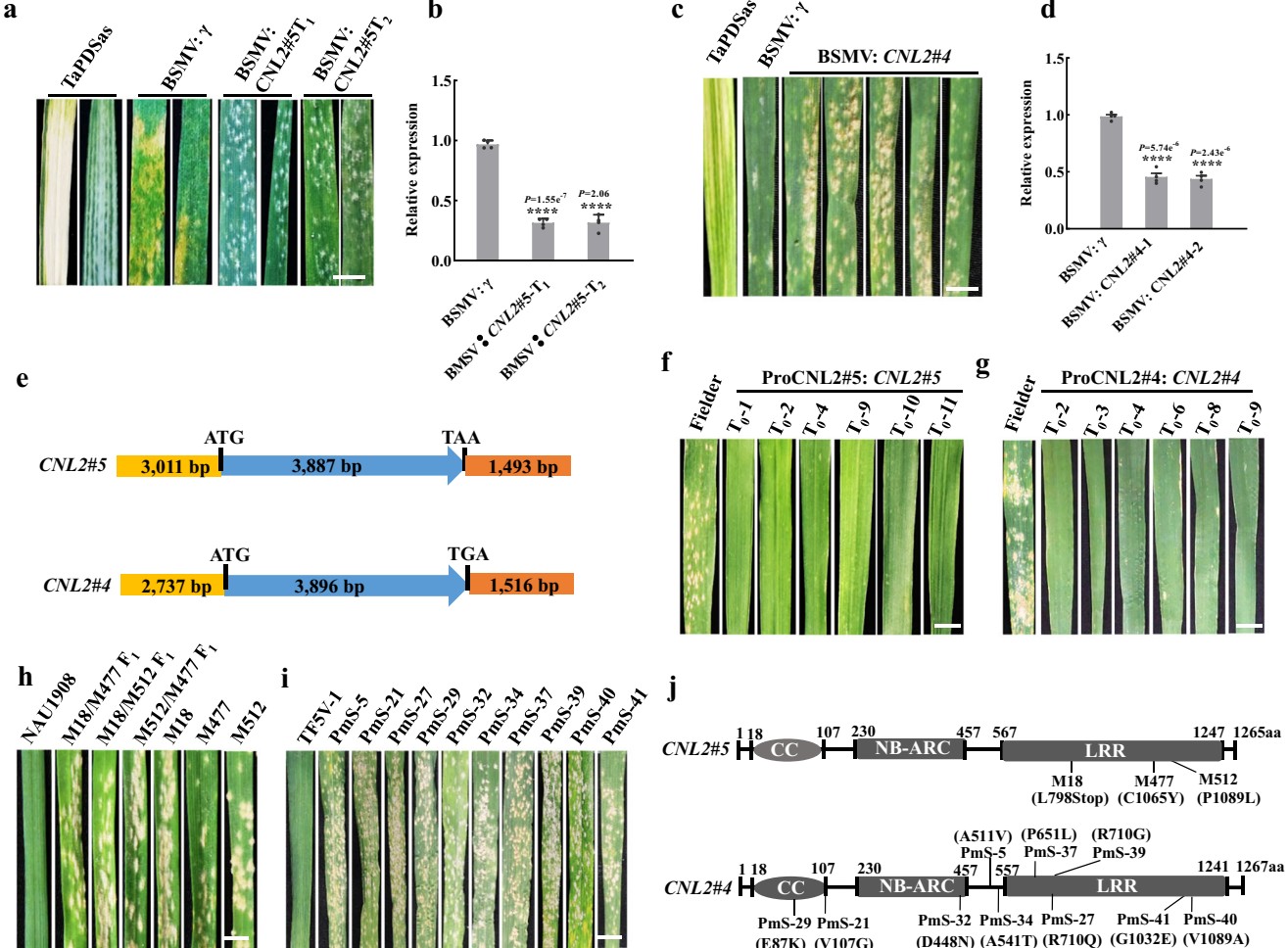

**Fig. 3 | The function validation of alleles *CNL2#4* and *CNL2#5*. a** Targets of CNL2#5 in NAU1908 silenced by VIGS exhibiting susceptibility on seedling leaves. Scale bar, 5.0 mm. **b** Reduced expression levels of *CNL2#5* in silenced leaves. Values are the mean ± SD (two-sided *t*-test, *n* = 4 biologically independent experiments, ****$P$ < 0.0001), BSMV: γ leaves were used as the control; **c** Target of *CNL2#4* in R5VS-15 silenced by VIGS exhibiting susceptible responses on seedling leaves. Scale bar, 5.0 mm. **d** Reduced expression levels of *CNL2#4* in silenced leaves. BSMV: CNL2#4-1 and BSMV: CNL2#4-2 are two independent injection experiments. Values are the mean ± SD of qRT-PCR (two-sided *t*-test, *n* = 4 biologically independent experiments, ****$P$ < 0.0001), BSMV: γ leaves were used as the control. **e** Structures of ProCNL2#5: CNL2#5 and ProCNL2#4: CNL2#4 used for transgenic assays. The

ProCNL2#5: CNL2#5 construct contained the entire *CNL2#5* gene, 3011 bp promoter, and 1493 bp terminal region. The ProCNL2#4: CNL2#4 construct contained the entire *CNL2#4* gene, 2737 bp promoter, and 1515 bp terminal region. **f** Responses of Fielder $T_0$ transgenic plants of ProCNL2#5: CNL2#5 to *Bgt* isolate E09. **g** Responses of Fielder $T_0$ transgenic plants of ProCNL2#4: CNL2#4. **h** Responses of *CNL2#5* EMS mutants and their $F_1$ plants. At least 10 plants of each lines were observed. Scale bar, 5.0 mm. **i** Responses of ten *CNL2#4* EMS mutants. At least 10 plants of each lines were observed. Scale bar, 5.0 mm. **j** EMS mutants carrying single nonsense or missense mutations in the *CNL2#4* and *CNL2#5* sequences. Black lines indicate the positions of the mutations. Source data are provided as a Source Data file.

sequence identity of >93% but lower than that of the *Pm2* series (>97%)[28]. A phylogenetic analysis clustered all predicted proteins to a single branch, consisting of sub-clusters of PM2 and PM55 homologs. All PM2 homologs clustered together phylogenetically, as did all PM55 haplotypes (Supplementary Fig. 17). Nonsynonymous (Ka) and synonymous (Ks) nucleotide substitution rates among PM55 full-length proteins were determined and a relatively lower Ka/Ks ratio (<1.0) indicated purifying selection (Supplementary Table 9).

### *SuPm55* interacts with *Pm55* alleles at protein level but does not suppress *Pm55b*-mediated resistance

The expression analysis indicated that *SuPm55* was highly expressed in non-infected TF5V-1 seedlings, but was significantly lower after the 3-leaf stage (Fig. 4a). Additionally, *SuPm55* was highly expressed in TF5V-1 leaf sheaths, with negligible expression in other organs at the adult stage (Fig. 4b). These observations suggested that *SuPm55* expression was developmentally regulated. We also detected the significant reduction of *SuPm55* expression in Del5VS-4 seedlings using

the qPCR primer that was located in the 27 bp deletion region (Supplementary Fig. 8a, Fig. 4c). However, the transcript levels of *Pm55a* in seedlings and adult-leaf sheaths were unaltered in TF5V-1 and Del5VS-4 plants (Fig. 4d), which indicated that *SuPm55* could not suppress *Pm55a* transcription. A probably of functional interference between *Pm55a* and *SuPm55* occurs at protein level.

We then conducted several assays to investigate the interaction between PM55 and SuPM55. Through Y2H assays, we found that PM55a and SuPM55 interacted through their CC domains, but not NB-ARC or LRR domains (Fig. 4e, Supplementary Fig. 18a). This interaction was further confirmed by co-immunoprecipitation (coIP), bimolecular fluorescence complementation (BiFC) and luciferase complementation (Luc) (Fig. 4f–h). Moreover, cell death assays demonstrated that expression of the *Pm55a* CC domain alone in *Nicotiana benthamiana* leaves induced hypersensitive response cell death in the injection region, whereas the longer CC-NBS or CC-NBS-LRR did not induce cell death responses (Supplementary Fig. 19a), suggesting that the full CC domain of *Pm55a* is active in

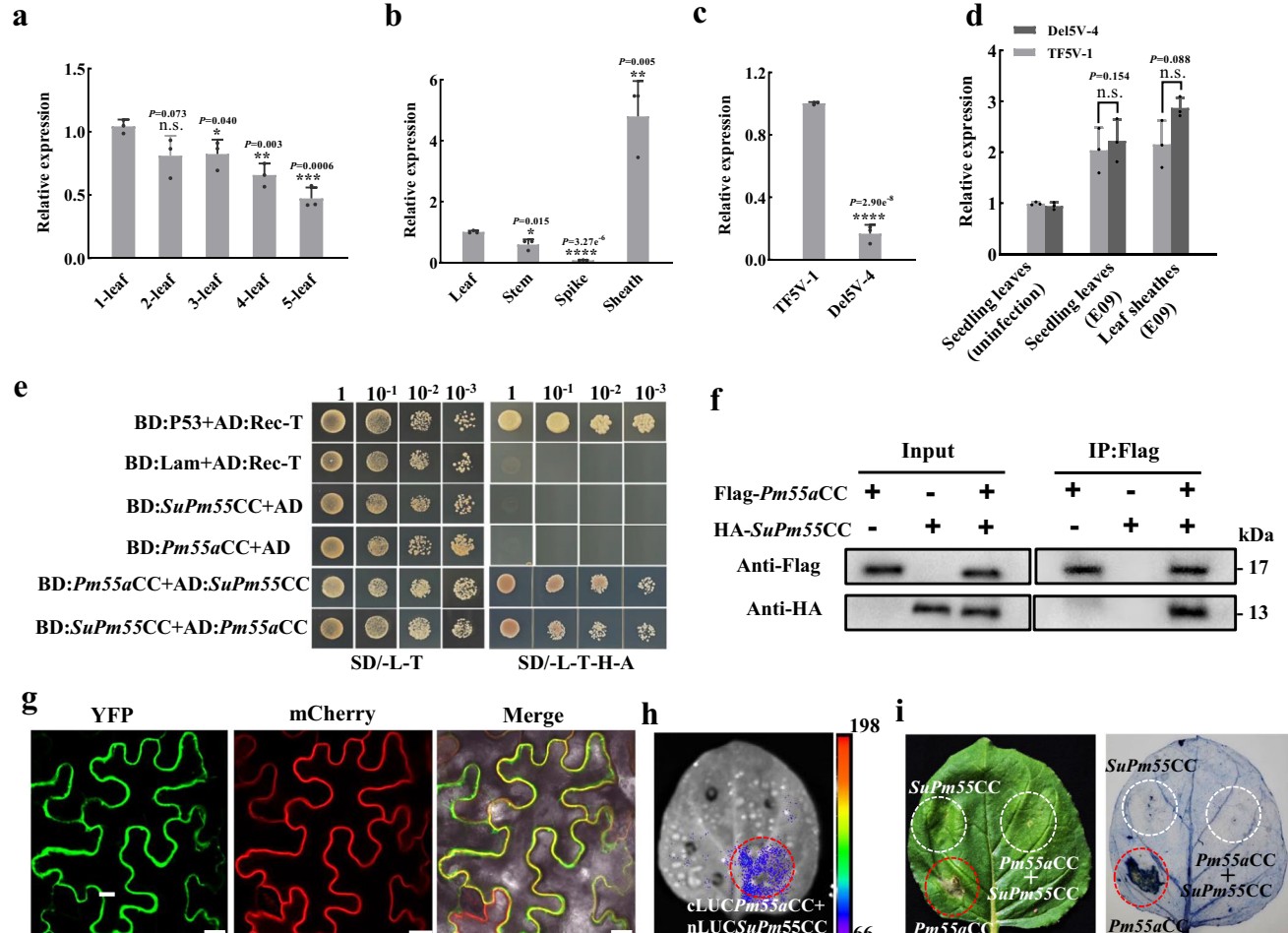

**Fig. 4 | The CC domains of *Pm55* and *SuPm55* interacted at protein level.**
**a** Expression levels of *SuPm55* in one-leaf to five-leaf seedlings. The values of qRT-PCR are the mean ± SD (two-sided *t*-test, *n* = 3 biologically independent experiments, ***P < 0.001, **P < 0.01, *P < 0.05, N.S., no significant). The one-leaf was used as control. **b** Expression levels of *SuPm55* in adult-plant tissues. The values of qRT-PCR are the mean ± SD (two-sided *t*-test, *n* = 3 biologically independent experiments, ****P < 0.0001, **P < 0.01, *P < 0.05). **c** Expression levels of *SuPm55* in leaf sheaths of TF5V-1 and Del5VS-4. The values of qRT-PCR are the mean ± SD (two-sided *t*-test, *n* = 3 biologically independent experiments, ****P < 0.0001). **d** Expression levels of *Pm55a* in TF5V-1 and Del5VS-4 at seedling and adult-plant stages. The values of qRT-PCR are the mean ± SD (two-sided *t*-test, *n* = 3 biologically independent experiments, N.S., no significant). **e** Interactions between the CC domains of *Pm55a* and *SuPm55* in Y2H assays. AD, activation domain; BD, binding domain. SD/-L-T, SD media lacking Leu and Trp; SD/-L/-T/-H/-A, SD medium lacking Leu, Trp, His and Ade. Numbers in column headings indicate dilution series of yeast cells expressing AD and BD fusions. Interaction between the CC domains of *Pm55a* and *SuPm55* detected by Co-immunoprecipitation (coIP) (**f**), BiFC (**g**), Scale bars, 20 μm, and Luc (**h**) assays. **i** The CC domain of *SuPm55* inhibited *Pm55a*-triggered cell death when their CC domains were both injected in *N. benthamiana* leaves. Source data are provided as a Source Data file.

cell-death signaling. However, the CC domain of *SuPm55* suppressed hypersensitive response cell death induced by *Pm55a* in tobacco leaves (Fig. 4i), indicating that *SuPm55* CC domain is essential to inhibit *Pm55a*-triggered cell death. Interestingly, expression of the *Pm55b* CC domain alone in tobacco leaves also induced hypersensitive response cell death (Supplementary Fig. 19b), and the CC domains of *Pm55b* and *SuPm55* interacted in Y2H assays (Supplementary Fig. 18b). However, the CC domain of *SuPm55* did not inhibit *Pm55b*-triggered cell death in tobacco leaves (Supplementary Fig. 19b), probably due to an amino acid change in the CC domain of *Pm55b* (Supplementary Fig. 7). Furthermore, type I recombinant lines (R5VS-27 to R5VS-31) that contained *SuPm55* and *Pm55b* showed no suppression in tests with *Bgt* isolates E09, E26 and E31 (Supplementary Fig. 3), as well as the R5VS-27 line showed similar response spectra as NAU1908 when tested with other 18 *Bgt* isolates (Supplementary Table 5). These findings collectively demonstrated that *SuPm55* did not suppress *Pm55b*-mediated resistance.

## Pyramiding *SuPm55*/*Pm55a* and *Pm55b* in wheat shows no suppression or yield penalty

We crossed the T5AL·5V#4 S translocation line NAU185 (*SuPm55*/*Pm55a*) with the T5DL·5 V#5 S translocation line NAU1908 (*Pm55b*) in order to combine the distinct resistance conferred by 5 V#4 S and 5 V#5 S. In the F₂ progeny, the homozygous multi-translocations line NAU2021 was identified using GISH/FISH (Supplementary Fig.20a, Fig. 5a,b). Detailed characterization indicated that NAU2021 has the similar development stages as the background parent NAU0686 (Fig. 5c), but displayed a reduction in plant height of approximately 5.0 cm compared to NAU0686 under field conditions without powdery mildew (Fig. 5d, e). However, NAU2021 did not show a significant reduction in yield-related trials, such as seeds per spike and thousand-grain weight (Fig. 5f, g), and instead showed a slight increase in spikes per plant and plot grain yield compared to NAU0686 (Fig. 5h, i).

To determine if the suppression of resistance could occur between allelic *Pm55a* and *Pm55b*, we initially tested NAU2021

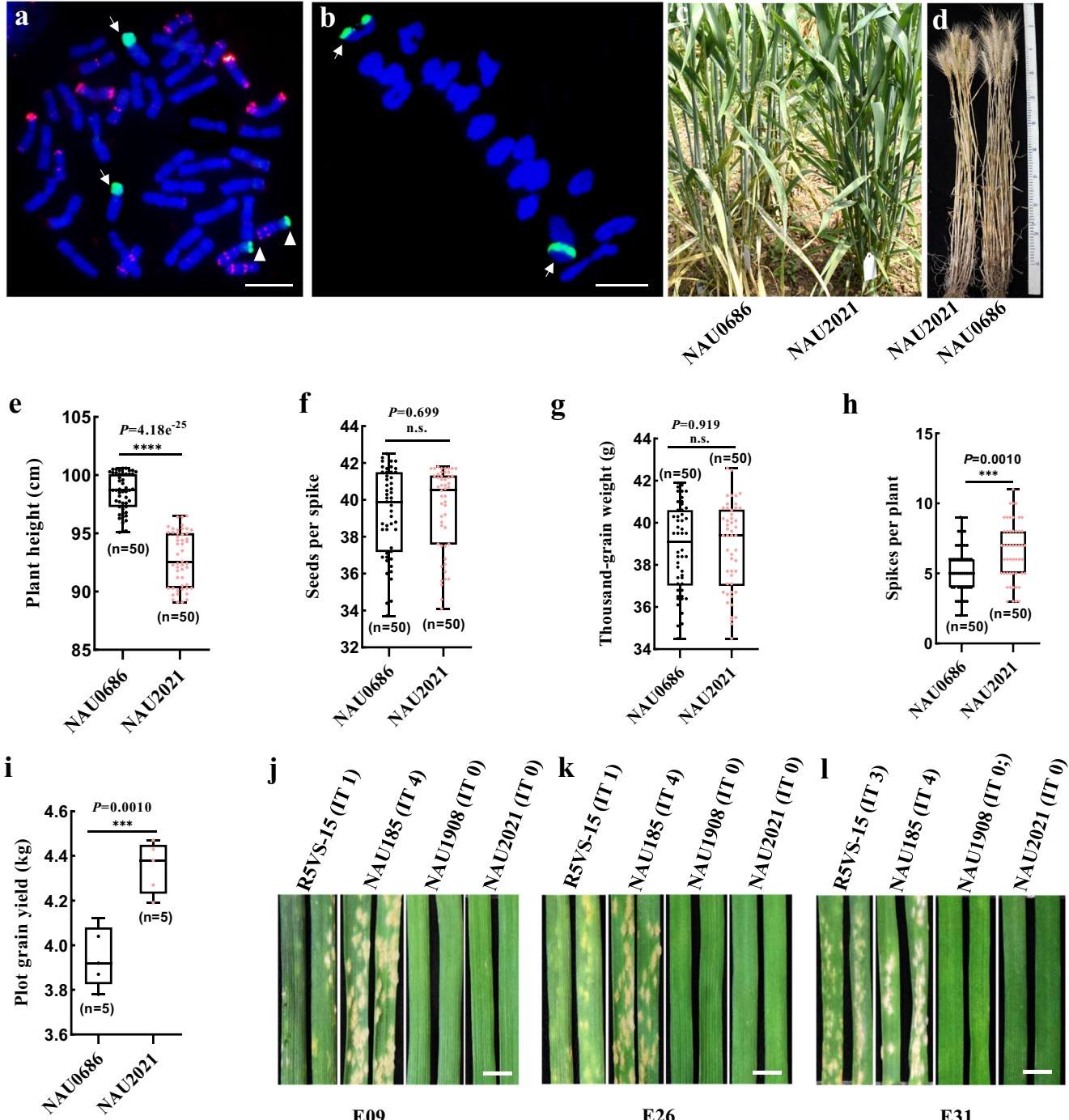

**Fig. 5 | Multiple 5 VS-translocations line NAU2021 shows no mutual allele suppression or yield penalty. a** Karyotypes of NAU2021 root-tip cell chromosomes. Arrows and triangles respectively indicate the translocated chromosomes T5AL·5V#4 S and T5DL·5 V#5 S. Bar, 10 µm. **b** Metaphase I of meiosis in NAU2021 pollen mother cell showing the 21 bivalents. Arrows indicate the translocated chromosomes pair forming two ring bivalents. Bar, 10 µm. **c**, **d** NAU2021 showing similar development stages as background parent NAU0686. Comparing the agronomic traits between NAU0686 and NAU2021 in plant height (*n* = 50) (**e**), seeds

per spike (*n* = 50) (**f**), TGW (*n* = 50) (**g**), spikes per plant (*n* = 50) (**h**) and plot grain yield (*n* = 5) (**i**). NAU2021 showing no yield penalty comparing with NAU0686, but significant reduction in plant height, and a little increase in spikes per plant and plot grain yield. Values are the mean ± SD (two-sided *t*-test, ****$P$ < 0.0001, ***$P$ < 0.001, N.S., no significant). The responses of NAU2021 seedlings tested with *Bgt* isolates E09 (**j**), E26 (**k**) and E31 (**l**). Ten plants of each lines were recorded the response to each *Bgt* isolates. Bars, 5.0 mm. Source data are provided as a Source Data file.

seedlings with *Bgt* isolates E09, E26 and E31. The results demonstrated that while *Pm55a* was susceptible to E31 (IT 4), NAU2021 seedlings did not compromise resistance to this isolate in *Pm55b* (Fig. 5j–l). Further, we compared the resistance spectrum of R5VS-15, NAU1908 and NAU2021 tested with the 18 *Bgt* isolates (Supplementary Table 5). The *Pm55a* line R5VS-15 showed susceptibility to three *Bgt* isolates 48-28 (IT

3), Nj-16 (IT 3) and E21-4 (IT 3), out of the 18 *Bgt* isolates. Whereas, *Pm55b* line NAU1908 exhibited medium resistance to isolates 48-18 (IT 2), 48-28 (IT 2) and Nj-16 (IT 2), and high resistance to the remaining isolates. By contrast, NAU2021 displayed additive resistance, as it showed medium resistance to isolates 48-28 (IT 2) and Nj-16 (IT 2), and immunity to the remaining 16 isolates. In addition, NAU2021 also

showed immunity to powdery mildew at the adult plant stage without a susceptible leaf sheath (Fig. 5c). Overall, the combination of T5AL·5V#4 S and T5DL·5 V#5 S translocated chromosomes did not lead to mutual allele suppression of resistance or yield penalty, thus providing a strategy to pyramid distinct resistance for durable and broad-spectrum resistance to powdery mildew in wheat (Supplementary Fig. 20b).

## Discussion

Alien introgressions have greatly enriched the otherwise limited gene pool in bread wheat. However, the successful map-based cloning of introgressed resistance genes has often been hampered by insufficient recombination in the alien chromatin regions[29]. Here, we employ T5DL·5 VS derivatives of different *D. villosum* accessions to develop the recombinants of chromosome arm 5VS. The identification of homologous recombinants of chromosome arm 5VS contributes to the evaluation of the complex responses to powdery mildew and ultimately confirms the resistance and suppression loci on 5VS. Importantly, our mapped-based cloning efforts were facilitated using chromosome flow sorting and sequencing 5V#4 S and 5 V#5 S DNA. The sequences obtained not only helped to develop polymorphism InDel markers for genetic mapping, but also facilitated in aligning their annotated genes with the reference genomes within the interested genome locus. The alignment of the annotated genes in the *SuPm55* interval with 5AS, 5BS, and 5DS of Chinese Spring revealed high collinearity, providing valuable insights into genetic variability and potential candidate genes for further analysis. Through the assembly of two 5 V#4 S scaffolds combined with two 5 V#5 S scaffolds, we obtained a contiguous sequence covering the *SuPm55* interval based on the high identity sequences that they shared. This led to the identification of six annotated genes as candidates for *SuPm55*. Notably, *SuPm55* (*CNL1*) is rare in *D. villosum*, and is absent in both 5 VS of 91C43^DH and 5V#5 S, suggesting that the 5 V#4 S flow-sorted sequence plays a key role in isolating *SuPm55*. While 5 VS short sequences did not cover the *Pm55* interval, the scaffolds containing upstream and downstream sequences of *Pm55* alleles are still valuable for functional analysis. Therefore, a combination of the reference genome and the flow-sorted target genome sequences represents a valid approach for cloning alien genes in wheat, especially for wild outcrossing species that usually results in complex haplotypes within the interested genetic locus.

The successful cloning of alien genes *Pm55*a and *Pm55*b adds to the repertoire of more than 10 NLR genes conferring powdery mildew resistance in wheat. Notably, *Pm55* is not orthologous to *Pm2* on chromosome arm 5 VS of *D. villosum*, which is distinct from other alien genes such as *Pm8*, an ortholog of wheat *Pm3*[30], and *Pm21*, an ortholog of wheat *Pm12*[31]. While the *Pm2* gene has been widely utilized in wheat breeding, it is currently losing its effectiveness against prevalent *Bgt* isolates in the main wheat production regions in China[32]. However, its ancestral orthologs, genetic mapping and expression analysis of *Pm2-5V#4* in TF5V-1 and *Pm2-5V#5* in NAU1908 revealed that they do not confer resistance. In contrast to highly conserved *Pm2* homologs in wheat and related genomes, the *Pm55* homologs exhibit significant sequence diversity and rapid evolution. The variation in amino acid composition between the PM55a and PM55b proteins is mainly located in the LRR domain, leading to a distinct resistance spectrum against *Bgt* isolates. Both *Pm55*a and *Pm55*b demonstrate broad-spectrum resistance powdery mildew based on infection assays with 21 *Bgt* isolates, highlighting their potential as resistance genes in wheat breeding.

Previous studies demonstrated that the CC domain heterodimerization of NLR proteins can abrogate disease resistance, as the homodimerization of NLR receptors via CC domains plays a role in R-mediated immunity[33]. For example, *PigmS* competitively attenuates paralogous *PigmR* homodimerization through heterodimers in their

CC domains to suppress the blast resistance in rice[4]. However, in case of homologous NLRs suppression among *Pm3* alleles, the N-terminal part of the LRR domain was identified as the major determinant of suppression[20]. In addition, *SuSr-D1*, a subunit of the mediator complex gene, has been shown to suppress stem rust resistance in wheat through regulating the expression of resistance genes[19]. Here, despite *Pm55* and *SuPm55* encode unrelated NLR proteins, the *SuPm55* CC domain can interact to suppress hypersensitive response cell death induced by *Pm55a*, but it does not inhibit *Pm55b*-triggered cell death in tobacco leaves. This suggests that the inhibition of triggered cell death may play a crucial role in the suppression of *Pm55a*-mediated resistance. Thus, the interactions of *Pm55* alleles with *SuPm55* might follow a different mechanism than that observed in interactions among homologous NLR immune receptors.

Plant NLR genes function as singletons, in pairs, or in networks in mediating resistance against pathogens[3]. Paired NLRs have been shown to regulate NLR-mediated resistance by balancing defense and fitness[11]. For examples, the rice NLR pairs *RGA4/RGA5* confer resistance to *Magnaporthe oryzae*[34], and *PigmS/PigmR* confer rice blast resistance[4]. In these cases, one of the NLRs triggers cell death but is constitutively expressed, while the other NLR represses its partner to prevent autoimmunity. Our results demonstrate that *Pm55a* is induced expression by *Bgt* and functions as singleton in transgenic plants, whereas the expression of antagonistic gene *SuPm55* is developmentally regulated by an unclear mechanism. In this case, introduction of *Pm55a* alone does not lead to autoimmunity in wheat, suggesting that the interaction between *Pm55a* and *SuPm55* differs from the paired NLRs. Interestingly, the inactivation of *SuPm55* significantly reduces grain yield, indicating that the expression of *SuPm55* in both seedlings and adult-leaf sheaths contributes to enhancing plant fitness. These results highlight that certain singleton NLRs should not be considered as individual entities, but rather as coevolutionary components with their inhibitors when utilized in breeding programs aimed at developing high-yield, efficient disease-resistance crops.

To breed crops with effective and durable resistance, it is highly desirable to combine different forms of resistance[10,35,36]. However, resistance alleles cannot be combined through classical crossbreeding. This limitation can only be overcome through genetic engineering or the use of F$_1$ hybrids[37]. In our study, we achieved the combination of distinct powdery mildew resistance conferred by *Pm55* alleles through T5AL·5V#4 S translocation with *SuPm55/Pm55a* and T5DL·5 V#5 S translocation with *Pm55b*. The T5AL·5 V#4 S and T5DL·5 V#5 S translocations line NAU2021, developed by the combination of classical crossbreeding, showed no mutual allele suppression or yield penalty in the field. Moreover, NAU2021 exhibited additional effective resistance when pyramiding the two translocated chromosomes in wheat. Therefore, the multi-translocations line NAU2021 represents a valuable resource for wheat improvement and provides a practical basis for pyramiding allelic series via crossbreeding to achieve a durable resistance.

## Methods

### Plant materials and field trials

*Dasypyrum villosum* accessions and two panels of wheat-*D. villosum* introgression lines listed in Supplementary Table 1 are maintained at the Cytogenetics Institute, Nanjing Agricultural University (CINAU). The powdery mildew susceptible wheat cv. Fielder was used for transgenic experiments. Yield trials of NAU0686 and NAU2021 were performed in randomized plot design with five replications. Each plot consisted of seven rows, measuring 3.5 m in length and 2.0 m in width. Fifty plants located in the middle of the internal rows of each materials were randomly selected for the analysis of yield-related traits, including plant height, thousand-kernel weight, spikes per plant, seeds spike and grain yield per plot.

### Bgt infection, staining and microscopy

Wheat *Bgt* isolates E09, E26 and E31 were provided by Prof. Yilin Zhou, Institute of Plant Protection, Chinese Academy of Agricultural Science, Beijing. Isolate E09 is virulent on *Pm1a, 3a, 3b, 3c, 3f, 3e, 5a, 6, 7, 8, 19, 25, 34* and *35*. Isolate E26 is virulent on *Pm1a, 3a, 3c, 3d, 3f, 3e, 5a, 6, 7, 8, 19* and *25*, whereas, E31 is virulent on *Pm1a, 2a, 3a, 3b, 3c, 3d, 3f, 3e, 4a, 4b, 5a, 6, 7, 8, 19, 24, 25, 30* and *34*. Additionally, the other 18 *Bgt* isolates collected in China were maintained at the Hubei Academy of Agricultural Science, at Wuhan. Powdery mildew responses on wheat seedlings were carried out in a greenhouse at 20–22 °C under a 12 h light/12 h darkness photoperiod. The *Bgt* isolates were maintained and increased on seedlings of wheat cv. Chancellor. Wheat lines were inoculated at the two-leaf stage and infection types (IT) were recorded on a 0–4 infection type (IT) scale at 7 days post inoculation when susceptible NAU0686 control plants were heavily diseased[13]. Adult-plant tests on lines grown in a field nursery were performed using the *Bgt* isolates E09. Susceptible cv. NAU0686 planted on both sides of each test row served as inoculum spreader and control. Reactions on leaves and sheaths were recorded at development stages of stem elongation, and heading on a 0–9 response scale[12].

To evaluate the differences in mycelial development, leaf segments were prepared using an endogenous peroxidase-dependent in situ histochemical staining procedure[13]. In order to detect leaf cell death, primary leaves from TF5V-1and NAU1908 plants were incubated by 0.5% (w/v) trypan blue staining at 7 days after *Bgt* isolate E09 inoculation (dpi)[38]. To visualize hydrogen peroxide ($H_2O_2$) accumulation, seedling leaves were stained at 2 dpi with 3,3'-diaminobenzidine (DAB) solution (1 mg/mL, pH 5.8) for 12 h, and then bleached in absolute ethanol[39]. The treated leaves were observed under an Olympus BX60 microscope (Olympus, Tokyo, Japan).

### Genome assembly of the PM55 and SuPM55 loci

To identify candidate genes in the genetic interval, we isolated and sequenced the translocated chromosome T5DL·5 V#4 S from line TF5V-1. Briefly, liquid suspensions of intact mitotic chromosomes were prepared from line TF5V-1. The chromosomes in suspension were fluorescently labeled by FISHIS using oligonucleotide 5'-FITC-GAA7-FITC-3' (Integrated DNA Technologies, Inc., Iowa, USA) and counterstained by DAPI (4´,6-diamidino 2-phenylindole). Next, bivariate flow karyotyping and chromosome sorting was done on a FACSAria II SORP flow cytometer and sorter (Becton Dickinson Immunocytometry Systems, San José, USA)[40,41]. The chromosome content of the flow-sorted fractions was determined by FISH on ~2000 chromosomes flow-sorted onto microscope slides using probes for pSc119.2, pTa71 and Afa family repetitive DNA sequences. Subsequently, DNA from flow-sorted chromosomes was purified, and the sheared DNA was used to prepare sequencing libraries using NEBNext Ultra™ II DNA Library Prep Kit for Illumina (New England Biolabs, Ipswich, USA). The libraries were sequenced on an Illumina NovaSeq 6000 to produce 2 × 250 bp paired-end reads. The size of the T5DL·5 V#4 S de novo assembly was 496.7 Mb with an average scaffold N50 size of 18.3 Kb. The total number of scaffolds was 57,258.

Moreover, a bacterial artificial chromosome (BAC) library was constructed using genomic DNA from *D. villosum* 01I140 (V#5). The nuclei of *D. villosum* accession 01I140 were isolated from approximately 30 g of etiolated young leaf tissue. High-molecular-weight (HMW) DNA was released from nuclei by proteinase K in lysis buffer (0.1 mg/mL proteinase K dissolved in 0.5 M EDTA, pH 9.1) at 50 °C for 48 h. The lysis buffer was changed after 24 h. Plugs (usually containing 5 to 6 μg undigested HMW DNA) were partially digested with *BamH I* or *Hind III*[42]. The BAC library comprised 351,520 clones and represented ~6.0-fold coverage of the *D. villosum* genome ( ~ 4.05 Gb). BAC clones were extracted from each of primary pools using a Qiagen Large-construct Kit (Germany). For each selected BAC clone, at least 300 x illumina paired-end short reads and 50x PacBio continuous long reads (CLR) were generated from the Illumina NovaSeq platform and PacBio Sequel platform, respectively. Library preparation and sequencing were performed at Novogene Co., Ltd (Beijing, China). Complete sequences for BAC clones were finally assembled from the short- and long- reads using an assembly workflow[43]. Finally, putative genes were annotated with the TriAnnot pipeline (https://urgi.versailles.inra.fr/triannot/?pipeline).

### Primer design and PCR amplification

DNA sequences from 5VS genome were used as templates for the development of molecular markers (Supplementary Table 10). All primers were designed using the Primer blast tool (*Triticeae* Multiomics Center/ViroBlast Home Page and DNAMAN V6). PCR amplifications were performed in 10 μL reaction mixture containing 5 μL 2 × Phanta® Master Mix (Vazyme, Nanjing, China), 0.2 μL of each primer, and 1 μL of genomic DNA. DNA amplification was performed at 95 °C for 3 min, followed by 33–35 cycles at 95 °C for 10 s, 55–60 °C (depending on the annealing temperature of primer pairs) for 30 s, and 72 °C for 30 s/Kb, with a final extension at 72 °C for 5 min. PCR products were separated in 1% agarose gels.

### Genomic DNA and RNA isolation and transcript analysis

Genomic DNA for molecular detection and gene cloning was extracted from seedling or adult-plant leaves of mutants, transgenic plants and parents by CTAB method[44]. Total RNA from the *Bgt*-inoculated leaves were extracted at 72 hpi for full-length transcript sequencing and cDNA analysis. Reverse transcription was performed using a HiScript III 1st Strand cDNA Synthesis Kit ( + gDNA wiper) (Vazyme, Nanjing, China). A cDNA library was constructed using SMARTer PCR cDNA Synthesis Kit (Tiangen, Nanjing, China), and sequenced using an Illumina platform at Beijing Biomark Biotechnology Co. Ltd. Raw data of 64.38 GB and 84.75 GB were obtained for NAU1908 and TF5V-1, respectively. To obtain a cyclic consensus sequence (CCS) the raw data were processed using pbccs 6.4.0 software provided by the PacBio Sequel platform. Isoseq3 3.8.2 software was used to remove chimeras and polyA to get full-length transcripts.

### Gene expression analysis by qRT-PCR

Total RNA was isolated using TriPure Isolation Reagent (Roche, Mannheim, Germany). First-strand cDNA was synthesized using 2 μg of total RNA by HiScript III reverse transcriptase (Vazyme Biotech, Nanjing, China) following the manufacturer's instructions. Quantitative reverse transcription polymerase chain reaction (qRT-PCR) was performed using SYBR Green (Vazyme Biotech, Nanjing, China) with a LightCycler® 480 instrument (Roche, Mannheim, Germany)[45]. The ACTIN gene was used as an internal control, and the $2^{(-\Delta\Delta CT)}$ method was used to calculate relative gene expression[46]. Three biological replicates and four technical replicates for each sample were performed.

### BSMV-induced gene silencing (BSMV-VIGS)

BSMV-VIGS was used to investigate candidate gene function in NAU1908, TF5V-1 and R5VS-15. A 211 bp fragment in the *CNL1* CC domain was inserted into the BSMV vector to form a recombinant virus BSMV: CNL1 was used to silence *CNL1* in TF5V-1. Fragments of 229 bp and 197 bp in the *CNL2#5* NB-ARC domain were separately inserted into the BSMV vector for silencing in NAU1908, and a 266 bp fragment in the *CNL2#4* NB-ARC domain for silencing in R5VS-15. A 265 bp fragment in the *Pm2-5V#4* LRR domain was use in silencing R5VS-15. The recombinant virus BSMV: PDS was used as a positive control and BSMV: γ was used as a negative control[47]. When the positive PDS control gene (phytoene dehydrogenase) was silenced and showed photobleaching symptoms, the fourth leaves of inoculated plants were taken for in vitro identification and determination of qRT-PCR silencing efficiency. Isolated leaves were cultured on 6-BA medium for 7 days, and disease development was assessed by hyphal development.

At least 10 plants of each genotype were challenged by each BSMV vector, and the experiments were repeated three times.

### EMS-induced mutants
Approximately 2000 seeds of NAU1908 and 3000 seeds of TF5V-1 were treated with 1.0% ethyl methanesulfonate (EMS). Treated seeds were then sown in the field to generate $M_1$ plants, and 678 independent $M_1$-generation plants of NAU1908 and 1042 $M_1$ plants of TF5V-1 were obtained, respectively. To screen susceptible mutants, about 50 seeds of each $M_1$ plant were evaluated for the response to $Bgt$ isolate E09. Susceptible $M_2$ plants were further advanced to the $M_3$ generation and their progeny tested. The full-length genomic sequences of $CNL2$ alleles were amplified from the susceptible mutants of TF5V-1 and NAU1908.

### Transformation $CNL2$ alleles
The 8391, and 8149 bp genome sequences encompassing the native promoters and terminators of $CNL2\#5$ and $CNL2\#4$, respectively, were cloned into the LGY-OE3 binary vector. The genomic fragments were inserted into the $Hind$ III−$Bam$ HI restriction endonuclease sites of the digested pLGY-OE3 with an In-Fusion HD Cloning Kit (Clontech Laboratories, Mountain View, CA, USA). The constructs were introduced into the powdery mildew-susceptible bread wheat cv. Fielder by $Agrobacterium$ $tumefaciens$-mediated transformation[48,49]. Both $T_0$ and $T_1$ plants were tested for the presence of the transgene by PCR amplification using the $CNL2\#4$- and $CNL2\#5$-specific markers, respectively. Wheat cv. Fielder was used as the negative control. The powdery mildew responses of $T_0$ transgenic plants and sibling controls were examined at the adult-plant stage, and $T_1$ and $T_2$ plants were test at the seedling stage as described above[50].

### CRISPR-Cas9 editing of $CNL1$
For editing $CNL1$ with CRISPR-Cas9, a sgRNA targeting the CC domain of $CNL1$ was designed. The Cas9-sgRNA expression vectors were constructed using CRISPRdirect (http://crispr.dbcls.jp/) and introduced into the $Agrobacterium$ $strain$ EHA105. $Agrobacterium$-mediated transformation was performed on immature embryos of TF5V-1[51]. Positive transgenic seedlings were selected with hygromycin (100 mg/L), and genomic DNA was extracted for PCR detection of the Cas9 gene fragment, 35 S promoter and hygromycin B phosphotransferase gene. PCR markers specific to $CNL1$ were used to identify the mutants in the CC domain. Homologous mutants were obtained by sequencing in $T_1$ and $T_2$ generations.

### Yeast two-hybrid (Y2H) assays
The MATCHMAKER GAL4 Two-Hybrid System 3 (Clontech) was used to examine the interaction between proteins. Appropriate amounts of bacterial solution were spread on SD/-Leu-Trp (SD/-L-T) media and incubated inverted at 30 °C for 3 days. The monoclonal antibody was picked up from SD/Leu-Trp-His-Ade (SD/-L-T-H-A) medium and resuspended in water, then diluted to a 1, $10^{-1}$, $10^{-2}$, $10^{-3}$ gradient, spotted on SD/-L-T-H-A medium, cultured at 30 °C and observed after one week[52].

### Bimolecular fluorescence complementation (BiFC) assays
The recombinant vectors Pm55-CC-nYFP and SuPm55-CC-cYFP were constructed using the CC domain sequences of $Pm55$ and $SuPm55$, and the $Agrobacterium$ GV3101 (Tiangen, Nanjing) containing the recombinant plasmid was inoculated in 20 mL LB liquid medium[53]. The cells were cultured at 28 °C until the logarithmic phase of $Agrobacterium$ growth, and were collected by centrifugation for 10 min at $5000 \times g$ and room temperature. $Agrobacterium$ was suspended by leaching solution (10 mM $MgCl_2$, 10 mM MES, 150 μM AS, pH = 5.6) to OD600 = 0.8–1.0, and then incubated for 3 h. Equal volumes of mixed bacterial suspension was injected into $N. benthamiana$ leaves, and after 48 h, the fluorescence signal of YFP was observed by laser confocal microscope and photographed (Leica SP8, Germany).

### Cell death assay in $N. benthamiana$ leaves
$Agrobacterium$ strain GV3101 carrying relevant plasmids was suspended in buffer to OD = 0.5, it was then injected into $N. benthamiana$ leaves, followed by the observation of cell necrosis for 24−72 h. After the appearance of the cell death, the leaves were stained with 0.4% TPN and subsequently decolorized with ethanol: acetic acid (3: 1, V/V) until the background became invisible. Finally, the leaves were photographed for recording[54].

### Luciferase complementation (Luc) imaging assay
To investigate interactions between the CC domains of $Pm55$ and $SuPm55$, a luciferase complementation assay was performed. In this assay, bacterial solution of cLUC-$Pm55$CC/nLUC-$SuPm55$CC, cLUC/nLUC-$SuPm55$CC, cLUC-$Pm55$CC/nLUC, and a mixture containing empty vectors cLUC and nLUC were injected into four different regions of the $N. benthamiana$ leaves[55]. The signals were detected 48 h after infiltration, and 10 leaves were analyzed. NightShade LB985 (Berthold Technologies, Germany) for fluorescence detection.

### Co-immunoprecipitation assay
Co-immunoprecipitation (Co-IP) experiments were performed following the Pierce HA Tag IP/Co-IP Kit instruction (Thermo). The transient expression of $N. benthamiana$ leaves was followed by Co-immunoprecipitation assay. The fusion protein carrier carrying the expression Flag- or HA- tag is transferred into $Agrobacterium$ strain GV3101, and the $Agrobacterium$ cells containing Flag-Pm55CC and HA-SuPm55CC are mixed and infiltrated into the leaves of $N. benthamiana$ at OD600 = 0.6 for full expression of the protein. Cracking bait-target cells and separation and purification of protein complexes, bait compound boiled and degeneration in SDS, then use 10% SDS-PAGE separation precipitation, with detection of HA or Flag antibodies (1:500, Abcam, Shanghai, China, No. AB 9110, AB 205606).

### Cloning and sequencing of $Pm55$ homologs
The primer pairs used for cloning full-length genomic sequences of $Pm55$ homologs are listed in Supplementary Table 10. Amplified fragments were ligated to a pToPo-Blunt (Aidlab, Nanjing, China) for sequencing at TongYong Co. (Nanjing, China). Putative domains of the cloned gene were analyzed using BLAST (http://www.ncbi.nlm.nih.gov/blast/). Protein prediction and multiple sequence alignment analysis were performed by the software of SMART (http://smart.embl-heidelberg.de/) and DNAMAN 7.0 software (Lynnon Biosoft, USA), respectively. R gene protein sequences with an N-terminal coiled-coil domain (CNL class) from the NCBI database were aligned using MUSCLE and a phylogenetic tree was constructed using the UPGMA (unweighted pair group method with arithmetic mean) program in MEGA 6.0 software[56]. Evolutionary distances were determined by the Neighbor-Joining method with Poisson correction, and the units were used to show the number of amino acid substitutions per site.

### Statistical analysis
The mean values and standard errors of the treatments were determined by Microsoft Excel. $T$ or ANOVA test was performed with SPSS 26.0 software (SPSS, Inc., Chicago, IL) to determine the significance of differences. Significant differences between two treatments were determined with a probability ($P$) value.

### Reporting summary
Further information on research design is available in the Nature Portfolio Reporting Summary linked to this article.

## Data availability

Data supporting the findings of this work are available within the paper and Supplementary Information files. The plant materials and datasets generated and analyzed during the present study are available from the corresponding authors upon request. Detailed genomic sequences of *Pm55* (OQ928403), *Pm5V* (ON109832), *SuPm5V* (OQ928410), *Pm2-5V#4* (OQ928409), *Pm2-5V#5* (OM646566), *Pm55_h1* (OQ928404), *Pm55_h2* (OQ928405) *Pm55_h3* (OQ928406), *Pm55_h4* (OQ928407), and *Pm55_h5* (OQ928408) were deposited in NCBI Genbank. The following public databases were used in this study: *D. villosum* 91C43^DH genome [https://bigd.big.ac.cn/], IWGSC RefSeq v2.1 [https://wheat-urgi.versailles.inra.fr/Seq], and Triticeae genomes [http://wheatomics.sdau.edu.cn/]. All primers used in this study are listed in Supplementary Table 10. Source data are provided with this paper.

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

## Acknowledgements

We thank Prof. Robert McIntosh, University of Sydney, for reviewing the manuscript. We also thank Zdeňka Dubská, Romana Šperková and Jitka Weiserová for preparation of chromosome samples for flow cytometry. This work was supported by the National Natural Science Foundation of China (32272062, 31971938); The Special Fund for Independent Innovation of Agricultural Science and Technology in Jiangsu (No. CX (19) 1001); and the "JBGS" Project of Seed Industry Revitalization in Jiangsu Province (JBGS (2021) 013). IM was supported from a Marie Curie Fellowship grant award 'AEGILWHEAT' (H2020-MSCA-IF-2016-746253) and from the Hungarian National Research, Development and Innovation Office (K135057), KH, MS and JD were supported from the ERDF project "Plants as a Tool for Sustainable Global Ddevelopment" (No. CZ.02.1.01/0.0/0.0/16_019/0000827). Computational resources were supplied by the project "e-Infrastruktura CZ" (e-INFRA LM2018140) provided within the program Projects of Large Research, Development and Innovations Infrastructures.

## Author contributions

R.Z. designed the study. C.L., J.D., H.C., S.G., Y.J., X.M., T.Z., B.F., I.M., K.H., M.S., L.X., L.K., J.D., G.L. and J.W. performed the research. R.Z., C.L. and J.D. analyzed the data. R.Z., P.C. and A.C. wrote the paper.

## Competing interests

The authors declare no competing interests.

## Additional information

**Peer review information** : *Nature Communications* thanks Beat Keller, and the other, anonymous, reviewer(s) for their contribution to the peer review of this work. A peer review file is available.

