## [Peer Review File · Nature Communications]

Wheat Pm55 alleles exhibit distinct interactions with an inhibitor to cause different powdery mildew resistanceREVIEWER COMMENTS

Reviewer #1 (Remarks to the Author):

This paper describes the molecular identification of three genes involved in wheat powdery mildew resistance after introgression from the wild relative of wheat, *Dasyphyrum villosum*. They all encode NLR type of immune receptors and were identified in two different D.v. accessions. Each of these two introgressions encodes a resistance gene (Pm55 or Pm5V), whereas SuPm55 is a specific suppressor for Pm55 only present in one introgression. Pm55 and Pm5V were found to be allelic, whereas SuPm55 encodes an unrelated protein. The two alleles show resistance at all stages, SuPm55 results in the loss of Pm55 resistance in seedlings and adult-leaf sheaths. Interestingly, loss of SuPm55 results in development changes, resulting in yield penalty. It remains unclear what is the molecular basis of this defect, as it seems that SuPm55 is not widespread in the gene pool. A combination of the two Pm55 alleles resulted in no yield penalty. The suppression mechanism was studied biochemically with protein interaction studies. This revealed that the CC domain of Pm55 and SuPm55 interact with each other, providing a mechanistic explanation of a possible suppression mechanism.

The results are of high interest and show the complex interactions of different NLR type of immune receptors in powdery mildew resistance. The work also describes a molecular mechanism for stage-specific resistance to wheat powdery mildew. Such stage specificity of resistance is frequently observed in wheat breeding, but it remained unexplained until now at the molecular level. The work presented here adds to our molecular understanding of powdery mildew resistance by providing three novel NLR types involved, and by studying suppression of gene activity.

There are several aspects in the manuscript which need to be addressed. I structure these problems here in aspects of scientific content vs. presentation.

Content:

L117: If there is recombination between the Pm55 and Pm5V genes, they are by definition not alleles (unless there is intragenic recombination). Thus, the obtained data and the interpretation are not in accordance and there is a need for clarification: What is the purpose of using the three isolates? What is their Avr/vir formula and what can be deduced by the phenotypes obtained by different isolates? Why do you conclude that they are likely alleles if there is recombination?

Figure 1e: the contiguous genome sequences at the genetic loci of interest are actually virtual sequences constructed by genome sequencing of two different accessions. How can this be justified? The authors must discuss why they think this can be done and is a valid approach for their purpose.

The VIGS results description is puzzling. CNL1 is SuPm55 and its silencing results in resistance as shown in figure 2a. L179 states: "... specific targeting of CNL1 expression by VIGS disrupted the compromise of Pm55-mediated resistance in seedling." This is unclear.

L283: What is meant with injected? According to Materials and Methods, *Agrobacterium* infiltration was used. Or did you inject purified protein? This is not clear.

Figure 3a: It is difficult to see the mildew growth after silencing. Better photographs should be presented.

There is a need for evaluation if the used silencing constructs are specific for the target sequences. As genome sequences are available for both wheat and D.v., this can be easily done.

What is CK in Figure 4d?

CoIP experiment, Figure 4f, IP:FLAG : It looks like the HA-SuPm55CC construct binds unspecifically to the anti-flag beads. Thus, the result of the coIP does not support the hypothesis of protein interaction.

There is a need to test the interaction of the CC domain of Pm5V with SuPm55. If protein interaction is at the basis of suppression, then the non-suppressed allele should encode a CC domain that is not interacting with SuPm55.

The phenotyping results for mildew resistance against different isolates must be more clearly presented for the combined genotypes of Pm55 and Pm5V. Is there additive resistance and does pyramiding work, based on careful selection of appropriate isolates to determine additive gene action at different growth stages?

Presentation:

There are many problems in language and a lack of clarity in the description of the work. The problems are at many different levels, from typos to the logical and precise description of the findings. I list here a selection of problems, but this is by far not complete. Thus, the manuscript must be thoroughly revised and, importantly, needs careful editing for English language. It is sometimes difficult to evaluate if the problems in the description of the work are caused by language problems or if there is a problem of logics

Title, abstract, Figure S22 and discussion: the term “dynamic resistance” is unclear. The authors should state clearly what is meant. The resistance observed is gene and stage-dependent, in addition there is a suppressor gene involved. This can not be summarized by the term “dynamic”, which is more confusing than helpful.

L85: It should be described in detail that two different accessions of D.v. were used for Pm55 vs. Pm5V introgressions.

The presentation is not clear enough on the allelic nature of Pm55 and Pm5V (line 94). It is also not clear why later in the manuscript (once allelism is demonstrated) the two genes are not named Pm55a and Pm55b. This would simplify the manuscript and be in accordance with other work on powdery mildew resistance (Pm1a,b,c... , Pm2a,b,c..., Pm3a,b,c,...).

Line 105ff: I would be clearer if the authors would focus on the Pm55 alleles only. It makes the manuscript unnecessarily complex to mention the other D.v. translocations with powdery mildew resistance genes, although they are not used and further described in the work here. In particular, the reference to Pm67 is not relevant in the results section. This also refers to data on Pm67 presented in supp figure 1.

Line 112: Rephrase sentence

Line 140ff: Please consider to first describe the isolation of the resistance genes, and not the suppressor. The current presentation is logically difficult to understand.

L189: attenuates: this is the only time in the manuscript where “attenuate” is used. Is the gene a suppressor, or does it only attenuate?

L196: The complementation result... Unclear.

L203: It is unclear what is meant with “each of the CNL2 alleles”

L355: Replace “inaction” throughout the manuscript.

Supplementary Table 7: correct spelling to: Landmark, ArinaLrFor

Unclear text must be rephrased. Here a few examples:

First sentence in abstract.

L70: NLR encoding locus that contains a single R gene... : rephrase

L72: replace “by far”

L99: ...but did not suppressed...

L129: To verify the third gene.... ?

L152: rephrase

Throughout the manuscript: replace sheathes by sheaths

Similarly, correct: mediated by mediated

Reviewer #2 (Remarks to the Author):

The paper describes a very thorough study on the isolation and characterisation of the Pm55 locus and Pm5V locus, both introgressed into wheat from *Dasypyrum villosum*. The outcomes are extremely interesting with an inhibitor NLR (SuPm55) linked in coupling with the Pm55 allele that explains the curious developmental phenotype associated with this resistance. The Pm5V gene is shown to be allelic to Pm55, but even though Pm5V interacts with the inhibitor in YTH assays it appears phenotypically immune to suppression of its resistance phenotype. In contrast, SuPm55 clearly suppresses Pm55 resistance when the inhibitor is present at sufficient levels.

I full support publication of this extensive and impressive study. However, not as the manuscript is currently written. A significant effort is required throughout the manuscript to improve the quality of the writing.

Not only does the writing quality need to be improved the authors need to make major efforts to improve the readability of the manuscript.

For example, in the early stages of the paper readers are informed “Pm55-introgressed T5DL-5VS or T5AL-5VS lines became effective to powdery mildew after

seedling stage and had susceptible lower leaf sheathes only at adult-plant stage whereas Pm5V-introgressed T5DL-5VS line showed broadly effective against Bgt races at all stages “. It is only by going to Supplementary Table 1 does it become entirely clear that two 5VS introgressions in these lines came from two different *D. villosum* accessions. This type of lack of descriptive clarity and accuracy of genotypes is a problem throughout the manuscript. Exacerbating this problem is that many lines and accessions with different genotypes are used throughout the study. I suggest it would be very helpful if the authors designated the Pm55 genotypes of each accession/line in brackets ie. SuPm55/Pm55, Pm5V and Pm55, throughout.

In addition, the authors should consider the following;

Separating Pm55 from SuPm55 results in pleiotropic effects such as fewer spikes, fewer tillers and reduced grain yield (Supp Figure 11). It is therefore surprising that line NAU2021 (SuPm55/Pm55 + Pm5V) shows a significant increase in spike number and yield. Given SuPm55 interacts with both Pm55 and Pm5V in YTH assays wouldn't the expectation be that there would be less SuPm55 available to interact with Pm55, meaning the pleiotropic effects would be more similar to those seen in Supp Fig. 11, rather than apparently the complete opposite effects?

Along the same lines, were pleiotropic effects observed in the transgenic CNL2#4 lines? Particularly T0-4 given it has expression 4x the control. As lines that had SuPm55 mutated showed reduced tillering and spike number which would also be expected in transgenic lines expressing Pm55 alone, given an absence of the inhibitor.

In Supp Figure 3 Pm55 lines (type II) show resistance in adult plant leaves to three isolates used. (Is this correct?? Its not particularly clear if this adult resistance extends to all 3 isolates). Yet Type III recombination events in which SuPm55 is separated from Pm55 show seedling resistance to only isolates E09 and E26 and not E31. If so why does SuPm55/Pm55 show adult leaf resistance to E31 but not seedling resistance.

Line 136 : “In summary, a single allele Pm5V in NAU1908 conferred ASR to powdery mildew, whereas allele Pm55 together with a linked inhibitor gene, SuPm55 in TF5V-1 conferred the adult-plant tissue-specific resistance.” Perhaps also add – “Separation of the linked inhibitor gene from Pm55 uncovered seedling resistance that was effective against isolates E09 and E26, but curiously not E26, even though T5AL-5V#4S provides APR to all three isolates”, if this in fact correct.

Supplemental Table 5 suggests that NAU2021, which contains Pm55 and Pm5V, shows greater resistance to isolate 48-18 than does lines containing either gene alone?

On a more minor note;

I am not sure if adult plant resistance is the correct term for T5AL.5V#4S or T5DL.5V#4S resistance as while they are seedling susceptible they are also susceptible in the lower sheaths of adult plants?? Previously it was described as a developmental-stage and tissue-specific powdery mildew resistance gene – Zhang 2022 DOI: 10.1007/s00122-016-2753-8

A simple figure showing the deduced genetic structure of the locus (maybe as part of Figure 1 or supp) showing a PmV5 allele, a SuPm55/Pm55 allele and type III Pm55 allele (no inhibitor) would be very useful.

Line 129 “All F1 plants were susceptible to isolate E09 and the F2 population segregated to 128 resistant and 394 susceptible plants” (and supp figure 5). Given you are dealing with developmental resistance phenotypes you cant call them plants - they are either seedlings or adult plants! Presumably they were seedlings. Please specify the developmental stage of all material described throughout.

Lines 111 - NAU1901 carrying gene Pm67, which conferred ASR but with powdery mildew development on adult plant leaves – how is that ASR then??

Line 75 – the wheat suppressor encoding the mediator complex could be mentioned here by Hiebert et al 2020.

Figure 1a needs much better explanation in the legend. What does each dot represent. What do the colours mean?

Figure 1b seems an odd place to include FISH/GISH images as they are not mentioned in the Figure legend.

Line 166- why was the sequence “similar” rather than identical? Accession 011140 (V5) was the donor?

Supp Fig2a looks a little odd for DAB staining as this usually causes some brown staining of uninfected cell walls. Perhaps it is because the focus is on the leaf surface rather than underlying tissue?

What were the markers used – please specify ie. KASP, SNP etc.

Line 158: “Gene expression analysis revealed that the CNL2 allele, but not the CNL3 and CNL4 alleles, was expressed in seedlings of NAU1908”. Not correct - all three were expressed, only CNL2 was induced (Fig S7a).

More information is needed in the legend of Figure 2b. Does each point on the graph represent expression in an independent leaf? If so knockdowns 1-4 all show one expression data point that was only marginally less than wild type. Did these leaves show symptoms?

Line 188 – Figure 2 should be referenced not Figure 4.

Supp Figure 9 – the amount of fungal growth is reduced on the leaves not the amount of spores I would think.

Line 196 – “The complementation result was confirmed”.... I suggest 1– “These results were confirmed...”.

Table S3 and S4 – show P values rather than X2 values.

Supp Figures 14 and 15 – Which T0 plants are these T1 plants derived from. High expressors or low expressors.

Line 276- NB-ARC

Line 282 – not sure “as expected” is the right expression.

It is surprising that the 27 bp deletion in SuPm55 causes reduced expression (Fig. 4c)?

Line 344- why do the authors conclude that the SuPm55/Pm55 could be more durable when Supp Table 5 shows there is more virulence for this gene than Pm5V?

Reviewer #3 (Remarks to the Author):

The Pm55 alleles reveal divergent interaction with a linked inhibitor causing dynamic powdery mildew resistance in wheat

The manuscript by Lu et al identified three CNL genes that account for the tissue- or developmental stage- specific resistance to powdery mildew in wheat. Those CNL genes are designated as Pm55, Pm5V and SuPm55, and SuPm55 suppresses Pm55-, but not Pm5V-mediated resistance specifically in seedlings and adult-leaf sheathes. The tissue-specific suppression of Pm55 by SuPm55 appears to alleviate the Pm55 mediated growth suppression, contributing plant fitness. This mechanism helps wheat to maintain the effective defense to Bgt, while it minimizes the adverse effects on plant growth by Pm55.

The mapping and phenotyping data together with the VIGS and CRISPR/Cas9-mediated genome editing validate the candidate gene as SuPm55. Similarly, transgenic lines, VIGS and a mutagenesis assay indicate that Pm55 and Pm5V confer the underlying Bgt resistance, although I would like to see the knockdown by VIGS or knockout by Crispr/Cas9 of Pm55 gene (also Pm5v) in Del5VS lines for the ultimate proof.

Interestingly, Pm55 and Pm5V are allelic variants with 91.5% identity, but only Pm55 is suppressed by SuPm55. The authors showed the physical interaction between the N-terminal CC domains of Pm55 and SuPm55 using Yeast two-hybrid (Y2H) assays, Luciferase complementation (Luc) imaging assay, and co-immunoprecipitation. Furthermore, the CC domain of SuPm55 inhibited Pm55-CC triggered cell death.

Finally, the authors demonstrated that transferring Pm55, Pm5V and SuPm55 genes into elite lines using a donor parent line carrying the three genes provides durable resistance without yield penalty.

I think that this manuscript provides an important insight into plant immunity and breeding. Therefore, this manuscript will interest a broad readership after a few minor revisions. I have listed a few concerns below.

1) "Suppression mechanism by SuPm55"

A weakness of this manuscript is Fig. 4. I'm not entirely convinced by the data that the interaction between the CC domains of SuPm5 and Pm55 explains the suppression mechanism.

The authors should include the CC domain of Pm5v in the experiments. For example, if the CC domain of Pm5V triggers cell death in *N. benthamiana*, one would expect that the CC domain of SuPm55 does not inhibit the Pm5V triggered cell death.

2) Pyramiding Pm55, Pm5V and SuPm55

It is not very clear to me how the multiple 5VS-translocation line NAU2021 was generated or found.

3) Line 110, this not an "opposite" pattern, I would say a "contrasting" pattern.

4) Supplementary Fig. 20, some Chinese? characters remain.

Response to the reviewer's comments:

Reviewer#1

This paper describes the molecular identification of three genes involved in wheat powdery mildew resistance after introgression from the wild relative of wheat, *Dasypyrum villosum*. They all encode NLR type of immune receptors and were identified in two different D.v. accessions. Each of these two introgressions encodes a resistance gene (Pm55 or Pm5V), whereas SuPm55 is a specific suppressor for Pm55 only present in one introgression. Pm55 and Pm5V were found to be allelic, whereas SuPm55 encodes an unrelated protein. The two alleles show resistance at all stages, SuPm55 results in the loss of Pm55 resistance in seedlings and adult-leaf sheaths. Interestingly, loss of SuPm55 results in development changes, resulting in yield penalty. It remains unclear what is the molecular basis of this defect, as it seems that SuPm55 is not widespread in the gene pool. A combination of the two Pm55 alleles resulted in no yield penalty. The suppression mechanism was studied biochemically with protein interaction studies. This revealed that the CC domain of Pm55 and SuPm55 interact with each other, providing a mechanistic explanation of a possible suppression mechanism.

The results are of high interest and show the complex interactions of different NLR type of immune receptors in powdery mildew resistance. The work also describes a molecular mechanism for stage-specific resistance to wheat powdery mildew. Such stage specificity of resistance is frequently observed in wheat breeding, but it remained unexplained until now at the molecular level. The work presented here adds to our molecular understanding of powdery mildew resistance by providing three novel NLR types involved, and by studying suppression of gene activity.

There are several aspects in the manuscript which need to be addressed. I structure these problems here in aspects of scientific content vs. presentation.

Content:

L117: If there is recombination between the *Pm55* and *Pm5V* genes, they are by definition not alleles (unless there is intragenic recombination). Thus, the obtained data and the interpretation are not in accordance and there is a need for clarification: What is the purpose of using the three isolates? What is their Avr/vir formula and what can be deduced by the phenotypes obtained by different isolates? Why do you conclude that they are likely alleles if there is recombination?

Response: Thanks for your valuable and helpful comments. To evaluate the 5VS-translocation lines in response to powdery mildew accurately, we have respectively transferred T5AL·5V#4S, T5DL·5V#4S and T5DL·5V#5S translocated chromosomes into a highly susceptible cv NAU0686 genetic background by backcrossing, and the developed lines NAU185 (T5AL·5V#4S), TF5V-1 (T5DL·5V#4S) and NAU1908 (T5DL·5V#5S) have similar genetic background. To isolate the resistance genes on chromosome arm 5VS, we previously constructed the genetic mapping population by the crossing of two T5DL·5VS translocation lines TF5V-1 and NAU1908. The F₂ segregation fitted a 3R:1S ratio at the seedling stage inoculation with E26. Thus, it is easy to map *Pm5V* on 5V#5S based on the seedling phenotypes. However, all plants showed resistance phenotype and none susceptible plants (similar as NAU0686) were observed. Therefore, it is difficult to conclude the allelism of *Pm5V* and *Pm55* according to the confusing phenotypes. Consequently, we hope to produce the recombinant phenotype by inoculation with *Bgt* isolates E09, E26 and E31 at seedling stage, as E31 is virulence on most resistance genes but is avirulence on *Pm5V* (see their Avr/vir formula in the Material section of manuscript). The 5VS polymorphic molecular markers detected the F₃ and F₄ homozygous recombination lines that contained different crossovers on 5VS. Based on their phenotypes, we divided the recombinants into three types. Subsequently, the second cross was made between TF5V-1 and the Type III recombination

line R5VS-15. According to their F₁ and F₂ phenotypes, we confirmed the suppression locus *SuPm55*. Therefore, if we use the single isolate E09 or E26, we cannot distinguish Type III from Type I and Type II lines, as well as cannot find the suppression locus. According to all recombinants showed resistance E09 at adult plant stage, Type I lines were resistance to E31 at seedling stage, and Type III lines were susceptible to it, we concluded that *Pm55* and *Pm5V* could be allelic and had different resistance spectrum.

Figure 1e: the contiguous genome sequences at the genetic loci of interest are actually virtual sequences constructed by genome sequencing of two different accessions. How can this be justified? The authors must discuss why they think this can be done and is a valid approach for their purpose.

Response: We are grateful for your suggestion. Alignment of the sequenced genome of *D. villosum* 91C43^{DH}, the flow-sorted 5V#4S and 5V#5S genomes sequences can be used to develop markers and construct contiguous genome sequences at the genetic loci of interest. Flow-sorting and sequencing of the chromosome arm is help to isolate the alien genes in wheat, as most of the wild relatives of wheat are outcrossing species and the aimed genes may be absent in the reference genome, such as *SuPm55* that is missing in the 91C43^{DH} genome. We discussed this approach in the “Discussion” section of revised manuscript.

The VIGS results description is puzzling. CNL1 is *SuPm55* and its silencing results in resistance as shown in figure 2a . L179 states: .. Specific targeting of CNL1 expression by VIGS disrupted the compromise of *Pm55*-mediated resistance in seedling.” This is unclear.

Response: We are very sorry for our unclear description. We have rewritten the sentences of the description of VIGS results to make it clear and easy understand.

L283: What is meant with injected? According to Materials and Methods, *Agrobacterium* infiltration

was used. Or did you inject purified protein? This is not clear.

Response: We are very sorry for our unclear description. We have rewritten the sentence in the revised manuscript to make it clear. Only *Agrobacterium* infiltration was used, not the purified protein.

Figure 3a: It is difficult to see the mildew growth after silencing. Better photographs should be presented.

Response: Thanks for your suggestion. The clear photographs have been presented in the revised manuscript.

There is a need for evaluation if the used silencing constructs are specific for the target sequences. As genome sequences are available for both wheat and D.v., this can be easily done.

Response: Thanks for your suggestion. Yes, the silencing constructs of VIGS should be specific for the target sequences. To evaluate the specificities, the primers sequences of VIGS are used to align to the sequence of silenced genes; also, their PCR production is sequenced to confirm the target sequences are correct. We provided photographs to show the specificities of silencing constructs in the Supplementary Fig. **12a, b**.

What is CK in Figure 4d?

Response: The CK is non-infected seedlings of TF5V-1 and Del5VS-4. We have make it clear in the revised manuscript.

CoIP experiment, Figure 4f, IP:FLAG : It looks like the HA-SuPm55CC construct binds unspecifically to the anti-flag beads. Thus, the result of the coIP does not support the hypothesis of protein interaction.

Response: We apologize for the incorrect photos in Fig.4f of the original manuscript. We provided Input photographs twice, but none of the IP:Flag photos. The correct photos of IP:Flag were provided in the

revised manuscript. The original pictures were provided in the Source Data file.

There is a need to test the interaction of the CC domain of Pm5V with SuPm55. If protein interaction is at the basis of suppression, then the non-suppressed allele should encode a CC domain that is not interacting with SuPm55.

Response: Thanks for your suggestion. We added the results of the interaction of the CC domains of *Pm5V* with *SuPm55* in the tobacco leaf in revised manuscript. The results were shown in the Supplementary Fig. 19b.

The phenotyping results for mildew resistance against different isolates must be more clearly presented for the combined genotypes of Pm55 and Pm5V. Is there additive resistance and does pyramiding work, based on careful selection of appropriate isolates to determine additive gene action at different growth stages?

Response: We are grateful for the suggestions. We described the resistance to different isolates clearly in the revised manuscript. Combined genotypes of Pm55 and Pm5V in NAU2021 showed better resistance and agronomy traits than that of single gene line. We speculate that pyramiding T5AL.5V#4S and T5DL.5V#5S translocated chromosomes in wheat could have the additional effective resistance. Perhaps, combination of T5AL.5V#4S and T5DL.5V#5S also pyramids the minor resistance genes that enhancing the resistance.

Presentation:

Title, abstract, Figure S22 and discussion: the term “dynamic resistance” is unclear. The authors should state clearly what is meant. The resistance observed is gene and stage-dependent, in addition there is a suppressor gene involved. This can not be summarized by the term “dynamic”, which is more confusing than helpful.

Response: We agree with your suggestion. We described the distinct resistance of different 5VS-introgression lines in details in the revised manuscript.

L85: It should be described in detail that two different accessions of D.v. were used for Pm55 vs. Pm5V introgressions.

Response: Thanks for your suggestion. We described in details of the two 5VS introgression lines in the revised manuscript.

The presentation is not clear enough on the allelic nature of Pm55 and Pm5V (line 94). It is also not clear why later in the manuscript (once allelism is demonstrated) the two genes are not named Pm55a and Pm55b. This would simplify the manuscript and be in accordance with other work on powdery mildew resistance (Pm1a,b,c... , Pm2a,b,c... , Pm3a,b,c...).

Response: We agree with your suggestion, and renamed *Pm55* and *Pm5V* as *Pm55a* and *Pm55b* in the revised manuscript.

Line 105ff: I would be clearer if the authors would focus on the Pm55 alleles only. It makes the manuscript unnecessarily complex to mention the other D.v. translocations with powdery mildew resistance genes, although they are not used and further described in the work here. In particular, the reference to Pm67 is not relevant in the results section. This also refers to data on Pm67 presented in supp figure 1.

Response: Thanks for your suggestion. We deleted the Pm67 information in the revised manuscript.

Line 112: Rephrase sentence

Response: We rewrote this sentence.

Line 140ff: Please consider to first describe the isolation of the resistance genes, and not the suppressor. The current presentation is logically difficult to understand.

Response: Thanks for your suggestion. We rewritten the genetic analysis and try to make the result clear in the revised manuscript.

L189: attenuates: this is the only time in the manuscript where “attenuate” is used. Is the gene a suppressor, or does it only attenuate?

Response: Thanks for your suggestion. We revised attenuate as suppression.

L196: The complementation result... Unclear.

Response: We rewrote this sentence to make the result clear.

L203: It is unclear what is meant with “each of the CNL2 alleles”

Response: We rewrote this sentence to make it clear.

L355: Replace “inaction” throughout the manuscript.

Response: We revised “inaction” as “inactivation” throughout the manuscript.

Supplementary Table 7: correct spelling to: Landmark, ArinaLrFor

Response: We are very sorry for our incorrect writing, and have made them in correction.

Unclear text must be rephrased. Here a few examples:

First sentence in abstract.

L70: NLR encoding locus that contains a single R gene... : rephrase

L72: replace “by far”

L99: ...but did not suppressed...

L129: To verify the third gene.... ?

L152: rephrase

Throughout the manuscript: replace sheathes by sheaths Similarly, correct: madiated by mediated.

Response: Thank you so much for you carefully check. All the sentences and words mentioned above

were revised according to the comments.

Special thanks for your good comments!

Reviewer #2 :

The paper describes a very thorough study on the isolation and characterization of the Pm55 locus and Pm5V locus, both introgressed into wheat from *Dasypyrum villosum*. The outcomes are extremely interesting with an inhibitor NLR (SuPm55) linked in coupling with the Pm55 allele that explains the curious developmental phenotype associated with this resistance. The Pm5V gene is shown to be allelic to Pm55, but even though Pm5V interacts with the inhibitor in YTH assays it appears phenotypically immune to suppression of its resistance phenotype. In contrast, SuPm55 clearly suppresses Pm55 resistance when the inhibitor is present at sufficient levels.

I fully support publication of this extensive and impressive study. However, not as the manuscript is currently written. A significant effort is required throughout the manuscript to improve the quality of the writing.

For example, in the early stages of the paper readers are informed “Pm55-introgressed T5DL·5VS or T5AL·5VS lines became effective to powdery mildew after seedling stage and had susceptible lower leaf sheaths only at adult-plant stage whereas Pm5V-introgressed T5DL·5VS line showed broadly effective against Bgt races at all stages “. It is only by going to Supplementary Table 1 does it become entirely clear that two 5VS introgressions in these lines came from two different *D. villosum* accessions. This type of lack of descriptive clarity and accuracy of genotypes is a problem throughout the manuscript. Exacerbating this problem is that many lines and accessions with different genotypes are used throughout the study. I suggest it would be very helpful if the authors designated the Pm55 genotypes of each accession/line in brackets ie. SuPm55/Pm55, Pm5V and

Pm55, throughout.

Response: We are greatly thank for your valuable suggestions. We described the background of 5VS-introgression lines in details in the “Introduction” section of the revised manuscript. In addition, genotypes of each accession/line was marked in Supplementary Fig. 2 and manuscript to make the genotypes clear of the lines.

In addition, the authors should consider the following;

Separating Pm55 from SuPm55 results in pleiotropic effects such as fewer spikes, fewer tillers and reduced grain yield (Supp Figure 11). It is therefore surprising that line NAU2021 (SuPm55/Pm55 + Pm5V) shows a significant increase in spike number and yield. Given SuPm55 interacts with both Pm55 and Pm5V in YTH assays wouldn't the expectation be that there would be less SuPm55 available to interact with Pm55, meaning the pleiotropic effects would be more similar to those seen in Supp Fig. 11, rather than apparently the complete opposite effects?

Response: Thanks for your valuable comments. The inactivation of SuPm55 significantly reduces grain yield, indicating that expression of SuPm55 in both seedlings and adult-leaf sheaths contributes to enhancing the plant fitness. Thus, SuPm55/Pm55 should be used together in breeding. The line NAU2021, containing the translocated chromosomes of T5AL.5V#4S and T5DL.5V#5S showed a significant increase in spike number and grain yield than that of the background parent NAU0686. These results indicated that the other genes of agronomy traits-related might be present on the chromosome arm 5VS, or that the subgenome hybrid vigor of 5V#4S, 5BS and 5V#5S has effect on the agronomy traits. Previously studies have demonstrated that alien chromosome arm 5VS introgressed into divergent backgrounds of wheat has effect on agronomy traits, including increasing in spike number, reducing in plant height, drought tolerant, and grain softness. Supp Fig. 11 showed that the inactivation of SuPm55

significantly reduces grain yield, while Fig.5 showed that NAU2021 has positive effect on the grain yield.

They are not contradiction.

Along the same lines, were pleiotropic effects observed in the transgenic CNL2#4 lines? Particularly T0-4 given it has expression 4x the control. As lines that had SuPm55 mutated showed reduced tillering and spike number which would also be expected in transgenic lines expressing Pm55 alone, given an absence of the inhibitor.

Response: We are greatly thank for your valuable suggestions. Comparing transgenic lines with the receptor line cv Fielder that could conclude the pleiotropic effects of the genes CNL2#4 and CNL2#5. However, the lines of NAU185, TF5V-1 and NAU1908 contained the whole chromosome arm 5VS, but not only gene CNL2#4 or CNL2#5. The genetic effect of 5VS introgressed into some bread wheat varieties have been shown no negative effect on major yield-related traits. The inactivation of SuPm55 in T5DL.5V#4S line TF5V-1 showed lower spike number per plant, indicating that expression of SuPm55 in both seedlings and adult-leaf sheaths contributes to enhancing the plant fitness. While, we cannot conclude that the gene Pm55 has native or positive effect on the plant fitness. The observation of SuPm55 pleiotropic effects is very interesting and we are going to further study on the molecular mechanisms. Thus, we did not evaluate the agronomy traits in the transgenic CNL2#4 or CNL2#5 lines.

In Supp Figure 3 Pm55 lines (type II) show resistance in adult plant leaves to three isolates used. (Is this correct?? Its not particularly clear if this adult resistance extends to all 3 isolates). Yet Type III recombination events in which SuPm55 is separated from Pm55 show seedling resistance to only isolates E09 and E26 and not E31. If so why does SuPm55/Pm55 show adult leaf resistance to E31 but not seedling resistance.

Response: Thanks for your comments. The recombination lines were only tested with E09 at adult plant

stage in the field condition. We think that TF5V-1 containing SuPm55/Pm55 should be susceptible to E31 at adult plant stage.

Line 136 : “In summary, a single allele Pm5V in NAU1908 conferred ASR to powdery mildew, whereas allele Pm55 together with a linked inhibitor gene, SuPm55 in TF5V-1 conferred the adult-plant tissue-specific resistance.” Perhaps also add – “Separation of the linked inhibitor gene from Pm55 uncovered seedling resistance that was effective against isolates E09 and E26, but curiously not E31, even though T5AL.5V#4S provides APR to all three isolates”, if this in fact correct.

Response: Thanks for your valuable comments. In fact, isolate E31 is avirulence to Pm5V (Pm55b), but virulence to Pm55 (Pm55a). As E31 is virulence to most Pm resistance genes, it is prohibited to infect materials in the field condition. In the present study, the recombination lines were only infected to E09 (avirulence to most Pm resistance genes) at adult plant stage in the field condition.

Supplemental Table 5 suggests that NAU2021, which contains Pm55 and Pm5V, shows greater resistance to isolate 48-18 than does lines containing either gene alone?

Response: Thank you for your nice suggestions. Not only isolate 48-18, NAU2021 also showed better agronomy traits, such as grain yield. Previously studies have demonstrated that alien chromosome arm 5VS introgressed into divergent backgrounds of wheat has effect on agronomy traits, including increasing in spike number, reducing in plant height, drought tolerant, and grain softness. Perhaps, combination of T5AL.5V#4S and T5DL.5V#5S also pyramids the minor resistance genes that provides additive resistance.

On a more minor note;

I am not sure if adult plant resistance is the correct term for T5AL.5V#4S or T5DL.5V#4S resistance as while they are seedling susceptible they are also susceptible in the lower sheaths of adult plants??

Previously it was described as a developmental-stage and tissue-specific powdery mildew resistance gene – Zhang 2022 DOI: 10.1007/s00122-016-2753-8

Response: Thanks for your suggestion. We described as a developmental-stage and tissue-specific resistance in the revised manuscript.

A simple figure showing the deduced genetic structure of the locus (maybe as part of Figure 1 or supp) showing a PmV5 allele, a SuPm55/Pm55 allele and type III Pm55 allele (no inhibitor) would be very useful.

Response: Thanks for your suggestion. We showed the genetic structure of the locus in the revised Supplementary Fig. 2.

Line 129 “All F1 plants were susceptible to isolate E09 and the F2 population segregated to 128 resistant and 394 susceptible plants” (and supp figure 5). Given you are dealing with developmental resistance phenotypes you cant call them plants - they are either seedlings or adult plants! Presumably they were seedlings. Please specify the developmental stage of all material described throughout.

Response: Thanks for you professional suggestion. We described the results in details in the revised manuscript.

Lines 111 - NAU1901 carrying gene Pm67, which conferred ASR but with powdery mildew development on adult plant leaves – how is that ASR then??

Response: According to the comments of Reviewer#1, we delete *Pm67* information in the revised manuscript.

Line 75 – the wheat suppressor encoding the mediator complex could be mentioned here by Hiebert et al 2020.

Response: Thanks for your suggestion. We mentioned it (ref 19) in the revised manuscript.

Figure 1a needs much better explanation in the legend. What does each dot represent. What do the colours mean?

Response: According to the comments of Reviewer#1, we delete Figure 1a in the revised manuscript.

Figure 1b seems an odd place to include FISH/GISH images as they are not mentioned in the Figure legend.

Response: Thanks for your suggestion. We provided the explication in the revised manuscript.

Line 166- why was the sequence “similar” rather than identical? Accession 011140 (V5) was the donor?

Response: Thank you for pointing out this problem. Yes, 011140 is the donor. Here should be use “identical”.

Supp Fig2a looks a little odd for DAB staining as this usually causes some brown staining of uninfected cell walls. Perhaps it is because the focus is on the leaf surface rather than underlying tissue?

Response: Thank you for pointing out this problem. We provided a clear picture to replace the odd one in the Supplementary Fig. 1.

What were the markers used – please specify ie. KASP, SNP etc.

Response: Thank you for pointing out this problem. We added the markers type as InDel.

Line 158: “Gene expression analysis revealed that the CNL2 allele, but not the CNL3 and CNL4 alleles, was expressed in seedlings of NAU1908”. Not correct - all three were expressed, only CNL2 was induced (Fig S7a).

Response: Thank you for pointing out this problem. We revised the description.

More information is needed in the legend of Figure 2b. Does each point on the graph represent expression in an independent leaf? If so knockdowns 1-4 all show one expression data point that was only marginally less than wild type. Did these leaves show symptoms?

Response: Thanks for your suggestion. We provided more information of the legend of Figure 2b. BSMV: CNL1-1 to 4 represents four independent knockdown leaves. Each point on the graph represents independent expression data. Each leaf has four independent expression data.

Line 188 – Figure 2 should be referenced not Figure 4.

Response: Thank you so much for you carefully check. We revised it.

Supp Figure 9 – the amount of fungal growth is reduced on the leaves not the amount of spores I would think.

Response: Thanks for your suggestion. The seedling leaves of four SuPm55 deletion lines created by CRISPR/Cas9 were used to observe the mycelial development. At least 10 points of each lines were taken the photos for count the micro-colony index. Results showed that the mount of fungal growth and spores are all less than that of receptor line TF5V-1. We revised the results description in the manuscript.

Line 196 – “The complementation result was confirmed”.... I suggest 1– “These results were confirmed...”.

Response: Thanks for your suggestion. We revised this sentence.

Table S3 and S4 – show P values rather than X2 values.

Response: Thanks for your suggestion. The P values were showed in the revised manuscript.

Supp Figures 14 and 15 – Which T0 plants are these T1 plants derived from. High expressors or low expressors.

Response: Thanks for your suggestion. The T1 plants of *Pm5V* were derived from T0-9, 10 and 11; T1

plants of *Pm55* were derived from T0-2, 3 and 4. All of them were high expressionors after inoculation with E09. Please see Supplementary Table 4.

Line 276- NB-ARC

Response: We are very sorry for our incorrect writing. We have revised.

Line 282 – not sure “as expected” is the right expression.

Response: We are very sorry for our incorrect writing. We have revised the sentence.

It is surprising that the 27 bp deletion in *SuPm55* causes reduced expression (Fig. 4c)?

Response: Thanks for your suggestion. We detected the significant reduction of *SuPm55* expression in *Del5VS-4* seedlings using the qPCR primer that was located in the 27 bp deletion region (see Supplementary Fig. 8a).

Line 344- why do the authors conclude that the *SuPm55/Pm55* could be more durable when Supp Table 5 shows there is more virulence for this gene than *Pm5V*?

Response: Thanks for your suggestion. We think the susceptibility of leaf sheath conferred by *SuPm55/Pm55* contributes to keep the virulent strains present at a low frequency in the existing pathogen populations, or decrease the selection pressure for virulent variants of the pathogen, thus the combination of *SuPm55/Pm55* could provide more durable resistance than single *Pm55*. It may be a controversial view, so we delete this view in the revised manuscript.

Response to reviewer#3

The manuscript by Lu et al identified three CNL genes that account for the tissue- or developmental stage- specific resistance to powdery mildew in wheat. Those CNL genes are designated as *Pm55*, *Pm5V* and *SuPm55*, and *SuPm55* suppresses *Pm55*-, but not *Pm5V*-mediated resistance specifically in seedlings and adult-leaf sheathes. The tissue-specific suppression of *Pm55* by *SuPm55* appears to alleviate the

Pm55 mediated growth suppression, contributing plant fitness. This mechanism helps wheat to maintain the effective defense to Bgt, while it minimizes the adverse effects on plant growth by Pm55. The mapping and phenotyping data together with the VIGS and CRISPR/Cas9-mediated genome editing validate the candidate gene as SuPm55. Similarly, transgenic lines, VIGS and a mutagenesis assay indicate that Pm55 and Pm5V confer the underlying Bgt resistance, although I would like to see the knockdown by VIGS or knockout by Crispr/Cas9 of Pm55 gene (also Pm5v) in Del5VS lines for the ultimate proof. Interestingly, Pm55 and Pm5V are allelic variants with 91.5% identity, but only Pm55 is suppressed by SuPm55. The authors showed the physical interaction between the N-terminal CC domains of Pm55 and SuPm55 using Yeast two-hybrid (Y2H) assays, Luciferase complementation (Luc) imaging assay, and co-immunoprecipitation. Furthermore, the CC domain of SuPm55 inhibited Pm55-CC triggered cell death. Finally, the authors demonstrated that transferring Pm55, Pm5V and SuPm55 genes into elite lines using a donor parent line carrying the three genes provides durable resistance without yield penalty. I think that this manuscript provides an important insight into plant immunity and breeding. Therefore, this manuscript will interest a broad readership after a few minor revisions. I have listed a few concerns below.

1) “Suppression mechanism by SuPm55”

A weakness of this manuscript is Fig. 4. I’m not entirely convinced by the data that the interaction between the CC domains of SuPm5 and Pm55 explains the suppression mechanism. The authors should include the CC domain of Pm5V in the experiments. For example, if the CC domain of Pm5V triggers cell death in *N. benthamiana*, one would expect that the CC domain of SuPm55 does not inhibit the Pm5V triggered cell death.

Response: Thanks for your nice suggestion. We added the results of the interaction of the CC domains

of *Pm5V* with *SuPm55* in the revised manuscript. Results showed that the CC domain of *SuPm55* did not inhibit *Pm55b*-triggered cell death in tobacco leaves (Supplementary Fig. 19b), probably due to an amino acid change in the CC domain of *Pm55b* (Supplementary Fig. 7).

2) Pyramiding Pm55, Pm5V and SuPm55

It is not very clear to me how the multiple 5VS-translocation line NAU2021 was generated or found.

Response: Thanks for your suggestion. A graphical representation in Supplementary Fig. 20a was provided to explain the generation of the NAU2021. We crossed the T5AL·5V#4S translocation line NAU185 (SuPm55/Pm55a) with the T5DL·5V#5S translocation line NAU1908 (Pm55b), in the F₂ progeny, the homozygous multi-translocations line NAU2021 was identified by using GISH/FISH.

3) Line 110, this not an “opposite” pattern, I would say a“contrasting” pattern.

Response: Thanks for your suggestion. We have changed "an opposite" as "a contrasting" in the resubmitted manuscript.

4) Supplementary Fig. 20, some Chinese? charters remain.

Response: Thank you so much for you carefully check. We have deleted the Chinese words in Supplementary Fig. 20.

We greatly appreciate all the efforts of the editor and the reviewers on our manuscript!

REVIEWER COMMENTS

Reviewer #1 (Remarks to the Author):

The authors have strongly improved the description of their work and the manuscript. They have addressed most of my points. There are two aspects that still need improvement:

Major:

Original comment:

Figure 1e: the contiguous genome sequences at the genetic loci of interest are actually virtual sequences constructed by genome sequencing of two different accessions. How can this be justified? The authors must discuss why they think this can be done and is a valid approach for their purpose.

This comment has been addressed by adding a description of the flow sorting method in the discussion. This is not sufficient and the original comment has to be addressed in the results. The combination of sequencing data of two different cultivars must be clearly justified. There can be highly divergent haplotypes in different accessions, and this must be addressed.

Minor:

L112 and L125: When recombinants are mentioned, it would be helpful to describe "recombinants between marker X and marker Y". Thus, the reader does not have to search for this information.

Reviewer #2 (Remarks to the Author):

It is very pleasing to see that the authors have included many of my suggestions in their amended manuscript and generally given satisfactory responses to my questions. In particular they have made the genetic descriptions of the material they used far more readily accessible to the reader throughout, which greatly enhances the understanding the manuscript. Some additional improvement in English expression is still required in some instances however this will likely be achieved during the publication process. I fully support publication of this work.

Reviewer #3 (Remarks to the Author):

The authors have satisfactorily addressed my comments.

Response to the reviewer's comments:

Reviewer#1

Major:

Original comment:

Figure 1e: the contiguous genome sequences at the genetic loci of interest are actually virtual sequences constructed by genome sequencing of two different accessions. How can this be justified?

The authors must discuss why they think this can be done and is a valid approach for their purpose.

This comment has been addressed by adding a description of the flow sorting method in the discussion. This is not sufficient and the original comment has to be addressed in the results. The combination of sequencing data of two different cultivars must be clearly justified. There can be highly divergent haplotypes in different accessions, and this must be addressed.

Response: *D. villosum* is an outcrossing diploid species that could result in complex haplotypes within the genetic locus of interest. Therefore, to isolate the genes on chromosome arms 5V#4S and 5V#5S, it is essential to obtain their sequences. Although *D. villosum* accession 91C43^{DH} was sequenced and a bacterial artificial chromosome (BAC) library using the DNA from *D. villosum* accession 011140 was constructed in our lab, we proceeded to flow-sorted and sequenced 5V#4S and 5V#5S genomes. The size of the 5V#4S *de novo* assembly was 171.8 Mb with an average scaffold N50 size of 18.3 kb, including 1,262 high-confidence protein-coding genes annotated by using protein homology-based prediction methods; whereas the size of the 5V#5S was 173.4 Mb with an average scaffold N50 size of 22.4 kb, annotating 1,350 high-confidence protein-coding genes. These sequences help not only in developing polymorphism InDel markers for genotyping in TF5V-1/NAU1908 F₂ population, but also in aligning their annotated genes with reference

genomes within the interested genetic loci. The homology alignment of the annotated genes in *SuPm55* interval with 5AS, 5BS and 5DS of Chinese Spring showed high collinearity. Subsequently, we identified two 5V#5S scaffolds, Scaffold4218 (20,202 bp) and Scaffold3806 (39,439 bp) and two 5V#4S scaffolds, Scaffold15749 (54,335 bp) and Scaffold38916 (46,667 bp) containing the homologs of the annotated genes in the inhibitor region of different genomes. Scaffold4218 on 5V#5S includes three annotated genes (G1, G2, and G3), while Scaffold15749 on 5V#4S includes two annotated genes (G3 and G4), both containing G3 alleles. Additionally, both Scaffold3806 on 5V#5S and Scaffold38916 on 5V#4S encompass G5 and G6 alleles. Whereas, 5V#4S scaffold Scaffold15749 and 5V#5S scaffold Scaffold3806 share a high identity DNA fragment (2,166/2,120, 94.82%). Consequently, these four scaffolds were assembled into a contiguous 100,963 bp sequence between markers *SCA4218* and *SCA39816*, covering the inhibitor interval and annotating six candidate genes. Unexpectedly, *SuPm55 (CNLI)* is rare in *D. villosum*, and is absent in both 5VS of 91C43^{DH} and 5V#5S. Thus, the 5V#4S scaffold Scaffold15749 proves to be essential in isolating *SuPm55 (CNLI)*. We added Figures (**Supplementary Fig. 5c and d**) to elucidate these results, and added some descriptions in the Result and Discussion sections. We are grateful for your professional suggestions, which have significantly improved our manuscript.

Minor:

L112 and L125: When recombinants are mentioned, it would be helpful to describe "recombinants between marker X and marker Y". Thus, the reader does not have to search for this information.

Response: Thanks for your suggestion. We added the markers when the recombinants described in the revised manuscript from L112 to L142 lines.

Reviewer #2 (Remarks to the Author):

It is very pleasing to see that the authors have included many of my suggestions in their amended manuscript and generally given satisfactory responses to my questions. In particular they have made the genetic descriptions of the material they used far more readily accessible to the reader throughout, which greatly enhances the understanding the manuscript. Some additional improvement in English expression is still required in some instances however this will likely be achieved during the publication process. I fully support publication of this work.

Response: We greatly appreciate your professional comments that help us to improve our manuscript greatly. We further improve the language of our manuscript try our best.

Reviewer #3 (Remarks to the Author):

The authors have satisfactorily addressed my comments.

Response: We greatly appreciate you for your efforts and professional comments on our manuscript!

REVIEWERS' COMMENTS

Reviewer #1 (Remarks to the Author):

The authors have addressed my two comments in a a very satisfactorial way. Thanks a lot.